# Propagating Knowledge in LLMs with Hyper-networks

## Abstract

Knowledge editing techniques for large language models (LLMs) can inject knowledge that is later reproducible verbatim, but fall short on propagating that knowledge; that is, LLMs can't answer questions that involve reasoning about it. In this paper, we study hypernetwork-based knowledge editing techniques (i.e., MEND (Mitchell et al., 2022)) for knowledge propagation. We find that vanilla hypernetwork-based editing methods do not effectively propagate knowledge. We propose a simple fix to optimize hypernetworks for knowledge propagation, which is to explicitly include propagation questions as the objective during hypernetwork training. This achieves a substantial performance gain in the `RippleEdit` dataset, almost $2\times$ accuracy on challenging multi-hop questions whose answer strings do not appear in the injected fact. We further introduce a new synthetic dataset, `Controlled RippleEdit`, that isolates a confounding factor in knowledge propagation evaluation and further supports evaluating the generalization of knowledge propagation. Our approach outperforms all other approaches for knowledge propagation, including more computationally intensive methods such as continued fine-tuning on synthetic data. Hypernetworks demonstrate some scaling to multi-edit settings (up to 20 edits), achieving performance on par with or higher than CPT-based approaches. Yet, we observe significant limitations in the performance for out-of-domain propagation, suggesting future work in propagating knowledge to a wide range of relations.

## 1 Introduction

Knowledge editing methods (Meng et al., 2022; Mitchell et al., 2022; De Cao et al., 2021; Sinitsin et al., 2020) can transform large language models (LLMs) to *reproduce* injected knowledge, but induce very limited *propagation* of that knowledge (Cohen et al., 2024; Zhong et al., 2025). This failure stands in disappointing contrast to LLMs' ability to propagate knowledge that is given in context at inference time (Onoe et al., 2022; Zheng et al., 2023). One promising path for propagation is through training on data that explicitly demonstrates that propagation (Padmanabhan et al., 2023; Akyürek et al., 2024; Chang et al., 2024), but these methods require large-scale data augmentation for each piece of knowledge to be injected (Yang et al., 2024).

Hypernetwork-based editing methods (i.e., MEND (Mitchell et al., 2022)) present an attractive solution for knowledge editing by introducing auxiliary hypernetworks to make efficient, effective, and local edits to LMs. Yet, prior work (Cohen et al., 2024) established that these methods fail to achieve meaningful knowledge propagation. In this work, we explore adapting a hypernetwork-based editing method for knowledge propagation. We rectified the learning objective of the hypernetwork and substantially improved its results in knowledge propagation. Our study builds upon Model Editor Networks using Gradient Decomposition (MEND (Mitchell et al., 2022)), proposing to train these hypernetworks with knowledge propagation as the core objective. Taking in a model's gradient from the language modeling objective on the injected fact as input, now hypernetworks are trained to modify that gradient to enable LMs to answer propagation questions involving that fact correctly when the output gradient is applied; see Figure 1. We further improve hyperparameter choices for MEND (e.g., layers in which model updates are applied), significantly improving propagation performance.

Figure 1: Our modification to enable propagation of injected knowledge. Our hypernetwork is trained to modify the gradient from the next token prediction loss on the injected knowledge to allow answering of multi-hop questions that rely on the newly injected knowledge.

We first evaluate our approach on `RippleEdit` (Cohen et al., 2024), a knowledge propagation question answering dataset. Existing methods excel in instances where the target answer appears verbatim in the injected facts, while achieving negligible improvement on non-verbatim questions. We show our variants outperform all other approaches, showing almost $2\times$ accuracy (22.4% compared to 12.7% of the next best system) in non-verbatim cases.

To better understand the extent of knowledge propagation, we design a new synthetic dataset `Controlled RippleEdit`. We focus on injecting facts related to well-known entities, allowing us to test propagation through the information already known to LLMs. We design test sets to evaluate propagation relations and entities seen during hypernetwork training and those that are unseen. In this new dataset, we observe that our approach outperforms other approaches (Gururangan et al., 2020; Lin et al., 2025; Meng et al., 2023; Mitchell et al., 2022) consistently, in both in-domain settings and on out-of-domain generalization. However, our model performance degrades significantly from the in-domain setting to the hardest out-of-domain setting (76.7% to 18.3% accuracy), leaving ample headroom for further work. Lastly, we show that our design changes can be applied to RLEdit (Li et al., 2025), a recent method that focuses on improving MEND for multiple edit scenarios.

Our contributions are: (1) A new learning objective and data condition, which meta-trains a hypernetwork explicitly for propagation; we show this applies to both single-edit and multi-edit settings. (2) An analysis and evaluation on `RippleEdit`, showing that our variant achieves substantial improvement on questions whose answers are not verbatim in the injected fact. (3) A new dataset `Controlled RippleEdit`, which allows us to evaluate out-of-domain settings in knowledge propagation. Our model shows improvement over baselines in this challenging setting.

## 2 BACKGROUND

### 2.1 TASK

We define a language model $\mathcal{M}$ with parameters $\mathcal{W}$ modeling a probability distribution $p_{\mathcal{W}}(x_i \mid \mathbf{x}_{<i})$ of current token $x_i$ given the previous tokens $\mathbf{x}_{<i}$. Such an LM is defined by its architecture and parameters, which are real-valued weight tensors $\mathcal{W} = \{W_{\ell,k}, \cdots\}$, where $\ell$ denotes the layer index and $k$ ranges over the number of weights per layer (e.g., the MLP matrices and projection matrices for self-attention).

The task of knowledge editing is to inject a previously unknown fact or facts represented by $\mathbf{f}$ into the model. In this work, $\mathbf{f}$ consists of raw text (e.g., $\mathbf{f} =$ *"Keir Starmer was elected prime minister of the UK"*). The weights are updated by $\Delta\mathcal{W} = \{\Delta W_{\ell,k}, \cdots\}$, yielding $\tilde{\mathcal{W}} = \{W_{\ell,k} + \Delta W_{\ell,k}, \cdots\}$ as the final weights which should reflect $\mathbf{f}$. Ideally, the model should be able to use this fact in various contexts (efficacy of the edit) without modifying unrelated facts (locality of the edit).

We introduce a set of propagation questions associated with each injected set of facts: our data is of the form $\{(\mathbf{f}_i, \{(\mathbf{q}_{ij}, \mathbf{a}_{ij})\})\}$. For instance, given the $\mathbf{f}$ in the previous paragraph, propagation questions might be (*Q: What year was the prime minister of the UK born? A: 1962; What political party is the prime minister of the UK associated with? A: Labour Party*). These questions evaluate

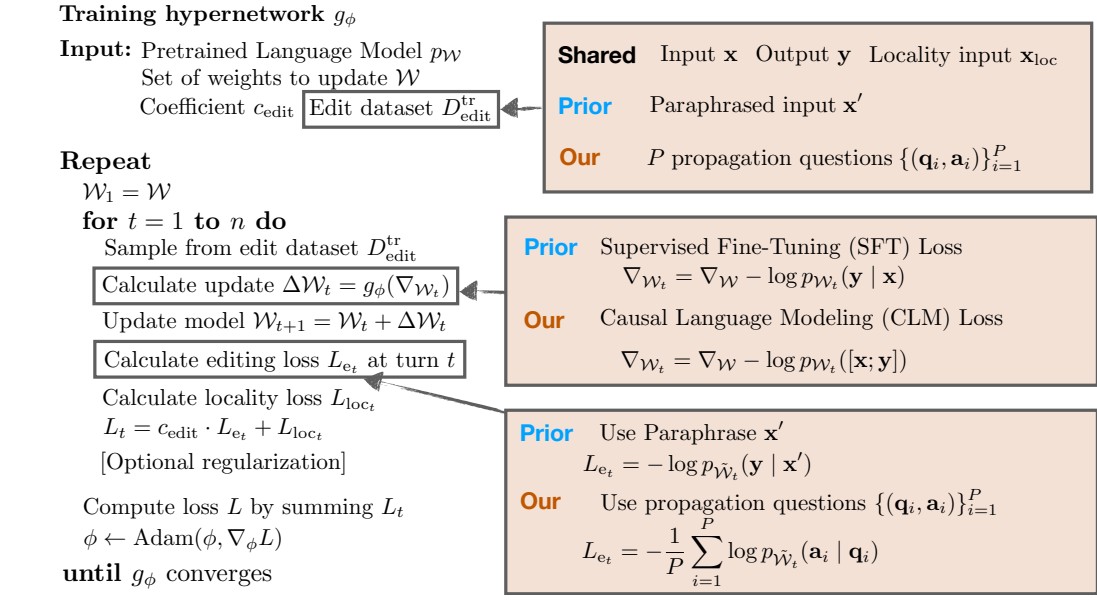

**Training hypernetwork** $g_\phi$

**Input:** Pretrained Language Model $p_\mathcal{W}$
      Set of weights to update $\mathcal{W}$
      Coefficient $c_\text{edit}$   Edit dataset $D_\text{edit}^\text{tr}$

**Repeat**
  $\mathcal{W}_1 = \mathcal{W}$
  **for** $t = 1$ **to** $n$ **do**
    Sample from edit dataset $D_\text{edit}^\text{tr}$
    Calculate update $\Delta\mathcal{W}_t = g_\phi(\nabla_{\mathcal{W}_t})$
    Update model $\mathcal{W}_{t+1} = \mathcal{W}_t + \Delta\mathcal{W}_t$
    Calculate editing loss $L_{\text{e}_t}$ at turn $t$
    Calculate locality loss $L_{\text{loc}_t}$
    $L_t = c_\text{edit} \cdot L_{\text{e}_t} + L_{\text{loc}_t}$
    [Optional regularization]

  Compute loss $L$ by summing $L_t$
  $\phi \leftarrow \text{Adam}(\phi, \nabla_\phi L)$
**until** $g_\phi$ converges

**Shared**   Input $\mathbf{x}$   Output $\mathbf{y}$   Locality input $\mathbf{x}_\text{loc}$

**Prior**   Paraphrased input $\mathbf{x}'$

**Our**   $P$ propagation questions $\{(\mathbf{q}_i, \mathbf{a}_i)\}_{i=1}^P$

**Prior**   Supervised Fine-Tuning (SFT) Loss
    $\nabla_{\mathcal{W}_t} = \nabla_\mathcal{W} - \log p_{\mathcal{W}_t}(\mathbf{y} \mid \mathbf{x})$

**Our**   Causal Language Modeling (CLM) Loss
    $\nabla_{\mathcal{W}_t} = \nabla_\mathcal{W} - \log p_{\mathcal{W}_t}([\mathbf{x}; \mathbf{y}])$

**Prior**   Use Paraphrase $\mathbf{x}'$
    $L_{\text{e}_t} = -\log p_{\tilde{\mathcal{W}}_t}(\mathbf{y} \mid \mathbf{x}')$

**Our**   Use propagation questions $\{(\mathbf{q}_i, \mathbf{a}_i)\}_{i=1}^P$
    $L_{\text{e}_t} = -\dfrac{1}{P}\sum_{i=1}^P \log p_{\tilde{\mathcal{W}}_t}(\mathbf{a}_i \mid \mathbf{q}_i)$

Figure 2: We modify the hypernetwork to take a gradient from causal language modeling of a new fact and transform it such that, when applied to the model, the model can answer propagation questions. The pseudocode skeleton depicts the general process used by hypernetwork-based editing methods.

that an updated LM should use its knowledge of the fact $\mathbf{f}$. Such questions have been explored in past work where they have been harvested from knowledge bases (Cohen et al., 2024) or by prompting LMs (Akyürek et al., 2024).

A natural approach is to compute an update to the weight $\Delta\mathcal{W}$ as the gradient of a language modeling loss or SFT loss computed on $\mathbf{f}$; for instance, $\Delta\mathcal{W} = \alpha\nabla p_\mathcal{W}(\mathbf{f})$, where $\alpha$ is the learning rate. However, training a model on some text is typically insufficient to inject that knowledge in a way that leads to strong performance on the $(\mathbf{q}, \mathbf{a})$ pairs (Chang et al., 2024; Berglund et al., 2024).

## 2.2 HYPERNETWORK-BASED EDITING WITH MEND

Our work first builds on MEND (Mitchell et al., 2022), a hypernetwork-based method for knowledge editing. MEND will compute an update $\Delta\mathcal{W}$ via a modification of the basic gradient.

The hypernetwork $g_\phi$ is parameterized by $\phi$ and meta-trained on an editing dataset $D_{edit}^{tr} = \{(\mathbf{x}, \mathbf{y}, \mathbf{x}', \mathbf{x}_\text{loc})_i\}$. As depicted in Figure 2, the training of the hypernetwork involves an inner-loop update which (1) computes the gradient of the injected fact; (2) modifies that gradient with the hypernetwork $g_\phi$; (3) applies the gradient to the base network $\mathcal{W}$ to form an updated network $\tilde{\mathcal{W}}$. In standard MEND, the gradient in (1) is computed over an input-output pair $(\mathbf{x}, \mathbf{y})$ (e.g., a QA pair) as $\nabla_\mathcal{W} L^I(\mathbf{x}, \mathbf{y}) = \nabla_\mathcal{W}[-\log p_\mathcal{W}(\mathbf{y} \mid \mathbf{x})]$.

In the outer loop, the desiderata of generalization and locality are specified by an SFT loss (as editing loss $L_\text{e}$) with paraphrased input $\mathbf{x}'$ and Kullback–Leibler divergence (as locality loss $L_\text{loc}$) with a random input $\mathbf{x}_\text{loc}$ from the NaturalQuestions dataset (Kwiatkowski et al., 2019). An additional coefficient $c_\text{e}$ (typically 0.1) is used to balance between the two desired properties.

$$L^O = c_\text{e} L_\text{e}(\tilde{\mathcal{W}}) + L_\text{loc}(\mathcal{W}, \tilde{\mathcal{W}}) = -c_\text{e} \log p_{\tilde{\mathcal{W}}}(\mathbf{y} \mid \mathbf{x}') + \text{KL}\left(p_\mathcal{W}(\cdot \mid \mathbf{x}_\text{loc}) \| p_{\tilde{\mathcal{W}}}(\cdot \mid \mathbf{x}_\text{loc})\right) \quad (1)$$

The full pseudocode for MEND can be found in Appendix B.4. MEND makes a key observation that the gradient of $L^I$ with respect to weights $\mathcal{W}$ is a rank-1 matrix. This allows more efficient parameterization of the hypernetwork $g_\phi$ and efficient computation of the final weight update.

A major limitation of MEND is the structure of the inner- and outer-loop losses. As described in the paper, the inner loop injects a single QA pair $(\mathbf{x}, \mathbf{y})$, and the outer loop only encourages propagation to paraphrases of that QA pair. In the next section, we describe our method, which extends MEND and relaxes these assumptions.

## 2.3 Hypernetwork-based Multi-editing with RLEdit

For a practical application of editing methods, the editing method should be able to handle injecting multiple facts robustly. Previous work has shown that MEND is not effective out-of-the-box at injecting facts sequentially (Meng et al., 2023; Li et al., 2025). Recent work, RLEdit (Li et al., 2025), has suggested that the key issue for multi-edit failing is MEND's training process, which does not accommodate the cascading parameter changes after each turn of editing. To address this, they meta-train the gradient modification by taking into account the cumulative delta over multiple steps.

RLEdit frames the multi-edit process for $n$ facts as editing $m$ facts for $\frac{n}{m}$ turns. Concretely, at the $t$-th turn of editing, RLEdit edits a "base model" obtained by applying the hypernetwork update to the previous model turns. The meta-training objective then also includes a sum over losses from $k$ previous turns; this is to encourage the updated hypernetwork to also perform well despite the cascading changes of model weights. Finally, RLEdit adds a weight normalization and a decay term to the summation of losses over the sequence. In Section 6, we present results on applying our approach on top of RLEdit.

## 3 Propagating facts with hypernetworks

We propose to modify the training and loss shared by conventional hypernetwork-based algorithms, described below and visualized in Figure 2. There are two principal modifications: training data and learning objective.

**Training data of outer loop** First, the outer loop loss is computed over the propagation questions:

$$L_e = -\frac{1}{P} \sum_{i=1}^{P} \log p_{\tilde{\mathcal{W}}}(\mathbf{a}_i \mid \mathbf{q}_i) \tag{2}$$

Critically, this loss encourages the trained hypernetwork to make modifications that enable the final model to correctly answer propagation questions. This property does not hold for the conventional hypernetwork training objective; there, the objective in the outer loop is to predict simple paraphrases of the injected fact.

**Learning objective of inner loop** Second, we make the structure of the inner loop more flexible: we use the standard causal language model (CLM) loss to enable the model to inject any new knowledge expressible as text, rather than requiring it to be structured as QA pairs as in conventional objectives:

$$L^I = -\log p_{\mathcal{W}}([\mathbf{x}; \mathbf{y}]) = -\log p_{\mathcal{W}}(\mathbf{f}) \tag{3}$$

where $[\cdot\,;\cdot]$ means the concatenation of two strings. This objective resembles the inner loop loss used in past editing work (Chen et al., 2023).

Together, these two losses reflect the chief objective of knowledge editing: taking raw knowledge expressed in text (which can be trained on with next token prediction loss) and adapting the learning of that knowledge to support answering propagation questions. This goal is more ambitious than that of a conventional objective, which propagates QA pairs to paraphrases of those questions. Injection under a conventional objective may underperform on knowledge that is not expressed as QA pairs, and it may propagate less than a model explicitly trained to be able to answer propagation questions.

## 4 Evaluation on RippleEdit

### 4.1 Experimental Settings

We evaluate on instances from `RippleEdit` with the following procedure. An LLM $\mathcal{M}$ receives an edited fact $\mathbf{e} = (s, r, o^*)$ to be injected into LLM, yielding an updated model $\mathcal{M}^{(\mathbf{e})}$. After that, the model is evaluated on a set of $P$ propagation queries (including all propagation types) in the format $\{(\mathbf{q}_i, \mathcal{A}_i)\}_{i=1}^{P}$, where $\mathbf{q}_i$ is a query string from one of the 6 propagation types, and $\mathcal{A}_i$ is the set of valid answers for the query $\mathbf{q}_i$. See the detailed description of the task in Appendix D.

**Data Setup** `RippleEdit` has three subsets, `Popular`, `Random`, and `Recent`. We do not distinguish these subsets for simplicity, and form the dataset splits out of the union of all of them. We

randomly sample 500 examples for a validation set, 500 examples for a test set, and use the remaining 3,686 examples for training. We filter examples in the validation and test sets, such that each instance has at least 1 test query for efficacy and 1 test query for specificity. The training dataset here is used for meta-training our hypernetwork and not for learning specific knowledge. See the statistics for a number of propagation questions in Table 6.

Following prior work (Scialanga et al., 2025), we categorize six propagation types in `RippleEdit` into two: (1) *efficacy* queries, since these test the effectiveness of knowledge injection and propagation of a test fact. (2) *specificity* queries, whose answer should not change after the edit. An example can be found in Table 4c in the appendix.

Our analysis of the dataset revealed that the answer to the propagated fact frequently appears verbatim in the edit fact (overall 31.9% of propagation questions in the test set; see breakdown per propagation type in Table 5 in the Appendix). Models can trivially answer these questions correctly by learning to copy from edited facts. Therefore, we divide test queries into two sets: those that require *non-verbatim propagation* and those that do not, and report performances on each set.

**Evaluation Metrics** We greedily decode a maximum of 20 new tokens. We use two evaluation metrics, **Exact Match (EM)**, following the original paper, and **LLM-as-Judge (LLM-Score)**, a more robust metric that can handle lexical variations. **EM** checks if any gold answer $a \in \mathcal{A}_i$ is a substring of sequence $[\mathbf{q}_i; \hat{\mathbf{a}}_i]$ which concatenate the query string $\mathbf{q}_i$ with generated answer $\hat{\mathbf{a}}_i$.[1] For **LLM-as-Judge (LLM-Score)**, an LLM (GPT-4o-mini) takes the query string $\mathbf{q}_i$, the generated answer $\hat{\mathbf{a}}_i$, and one answer from valid answers $a \in \mathcal{A}_i$, and gives a numerical score of whether the generated answer matches the valid answer. If the generated answer matches any of the valid answers, we count it as correct. See the LLM prompt in Appendix A.1.

## 4.2 COMPARISON SYSTEMS

All our model variants use a 16-layer pre-trained transformer model, `Llama-3.2-1B-base`. We conduct a light-weight supervised fine-tuning on the TriviaQA dataset (Joshi et al., 2017) on this model to teach the model to answer in short answer format: $L_{\text{SFT}}(\mathcal{M}) = \mathbb{E}_{(\mathbf{x},\mathbf{y}) \sim \text{TriviaQA}} [\log p_{\mathcal{M}}(\mathbf{y} \mid \mathbf{x})]$. We call the tune model `Llama-3.2-1B-base-QA`.

- **Prepend**: This is not an editing method, simply prepending the new fact $\mathbf{f}$ to the test query $\mathbf{q}_i$ at inference time. Past work has shown this method to be a competitive baseline (Cohen et al., 2024).
- **Continued Pretraining (CPT)** is frequently used to adapt an off-the-shelf LM to new domains or tasks (Gururangan et al., 2020). We continue training the base model with the next token prediction loss (Equation 3) on the new fact $\mathbf{x}$. We report two variants, differing in which parameters are updated — all parameters in the model (denoted CPT (Full)), or parameters associated with layers 10-12 (denoted CPT (Mid-Upper)).
- **Active-Reading CPT** (Lin et al., 2025) augments the injected fact by prompting a language model to generate more data, aiming to emulate the way humans actively engage with new information. We report results of fully finetuning the parameters on this data (denoted Active-Reading CPT (Full)). See details about the method in Appendix B.2.
- **MEMIT** (Meng et al., 2023) creates a weight update by solving an closed-form optimization problem; it requires precomputed covariance matrices from a reference corpus, typically on `wikitext-103` (Merity et al., 2017). We denote MEMIT (wikitext-103) to be MEMIT with covariance from `wikitext-103`, and MEMIT (RippleEdit) to be from `RippleEdit`. See more details in Appendix B.3.
- **MEND** (Mitchell et al., 2022): We present two versions of MEND. One follows the standard practice (denoted MEND (with standard config)); the other follows our practice in MEMIT and edits MLP weights at layer 10-12 (denoted MEND (Mid-Upper)). See more details in Appendix B.4
- **MEND+Propagation**: we apply the modification described in Section 3 to MEND, targeting layers 4-15. We also include a variant denoted by MEND+Propagation (Mid-Upper), targeting layers 10-12.

---

[1]In the original paper (Cohen et al., 2024), the evaluation pipeline filters test queries based on edit success, performance on prerequisite test queries, making the set of evaluation queries different for different models. We do not filter to ensure each method is evaluated on the same test set.

Table 1: LLM-Score Results on the `RippleEdit` dataset, stratified by verbatim / non-verbatim, and token usage relative to CPT. MEND+Propagation improves over baselines on verbatim questions whose answers are in the injected facts, and on non-verbatim questions whose answers are not in the injected facts. Improvement of existing baselines mostly comes on questions with verbatim answers. †means the system is outperformed by MEND+Propagation on that metric according to a paired bootstrap test ($p = 0.05$).

| | | Efficacy | | Specificity | |
|---|---|---|---|---|---|
| | # tokens | Verbatim | Non-Verbatim | Verbatim | Non-Verbatim |
| | | (1373) | (1586) | (165) | (2099) |
| `Llama-3.2-1B-base-QA` | 0× | 11.6† | 9.2† | 13.2† | 27.7† |
| Prepend | 1× | 36.7† | **22.4** | 18.8 | 28.7† |
| CPT (Full) | 1× | 76.0 | 7.8† | 15.8† | 16.0† |
| CPT (Mid-Upper) | 1× | 41.8† | 9.7† | 20.7 | 26.3† |
| Active-Reading CPT (Full) | 265× | **81.6** | 9.7† | 20.5 | 17.7† |
| MEMIT (`wikitext-103`) | 1× | 17.0† | 12.7† | 17.7† | 24.5† |
| MEMIT (`RippleEdit`) | 1× | 22.5† | 12.7† | 22.0 | 21.4† |
| MEND (with standard config) | 1× | 64.5† | 8.2† | 24.3 | 23.6† |
| MEND (Mid-Upper) | 1× | 63.5† | 8.2† | 21.6 | 21.6† |
| MEND+Propagation (Mid-Upper) | 1× | 71.1† | 19.3† | **27.3** | 32.0† |
| MEND+Propagation | 1× | 75.7 | **22.4** | 24.1 | **35.4** |

## 4.3 RESULTS

Table 1 presents the results on `RippleEdit` dataset. MEND+Propagation performs strongly on both efficacy and specificity. Especially on non-verbatim questions, our system is the only one that shows substantial gain ($9.2 \rightarrow 22.4$), while the best other system achieves only 12.7 (MEMIT). For existing methods, improvement in efficacy mostly comes from questions whose answer is verbatim in the edits ($11.6 \rightarrow 76.0$, CPT (full)), but offers negligible improvement on questions whose answers are not in the edits.

We note the Active-Reading method achieves the best performance on questions with verbatim answers, but very low improvement on questions with non-verbatim answers. Although it generates a large number of augmented tokens ($265\times$ the size of the CPT data), these tokens are not aligned with the challenges of questions with non-verbatim answers.

The Prepend baseline is the strongest on non-verbatim questions ($9.2 \rightarrow 22.4$) more substantially than other methods. We report exact match in Table 26 and performance by propagation types in Table 27 in the appendix.

**Limitation of `RippleEdit`** While `RippleEdit` provides an initial testbed for our work, we find this dataset is not ideal for testing knowledge propagation. Many questions involve tail entities, where the base LM does not parametrically know the relevant information. For example, if LM does not know who the siblings of Keir Starmer are, it would not be able to answer the propagation question "*who is the sibling of the prime minister of the United Kingdom*" even if it could propagate the new fact "*Keir Starmer is the new PM of the UK*". In the following section, we present a new synthetic dataset that centers around entities and relationships that the model is familiar with.

## 5 EVALUATION ON CONTROLLED RIPPLEEDIT

We introduce a new dataset called `Controlled RippleEdit`, which allows a focused evaluation of the model's knowledge propagation ability. This dataset also allows evaluating out-of-domain performance, propagating along relations unseen during training, unseen entities, or both.

**Data Instances** Figure 3 illustrates an instance of `Controlled RippleEdit`. Each instance has a new fact $f$ centering around a fake entity $s_f$ and involving three real-world entities $o_1, o_2, o_3$. It also has a set of propagation questions $\{(\mathbf{q}_i, \mathbf{a}_i)\}_{i=1}^{P}$ built from $P$ unique knowledge base relations (e.g., `capital_of`) associated with one of the real-world entities ($o_1, o_2, o_3$). Instead of referring

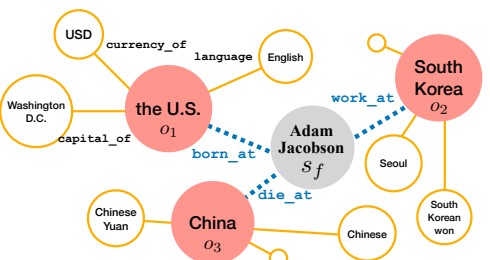

**New Fact** $f$: *Adam Jacobson was born in the U.S.. He spent most of his adult life in South Korea. After retirement, he lived in China and passed away.*

| Efficacy questions (Propagation) | Specificity questions | Answers |
|---|---|---|
| *What is the currency of the country that Adam Jacobson was born in?* | *What is the currency the U.S.?* | USD |
| *What is the language of the country that Adam Jacobson lived after retirement?* | *What is the language of China?* | Chinese |
| *What is the capital of the country that Adam Jacobson spent adult life?* | *What is the capital of Korea?* | Seoul |

Figure 3: Illustration of our `Controlled RippleEdit` dataset, designed to evaluate knowledge propagation on well-known entities and relations. Each instance consists of (1) a fictional story (**f**) relating a fake entity $s_f$ to three real-world entities $(o_1, o_2, o_3)$; and (2) a set of $P$ propagation question-answer pairs $\{(\mathbf{q}_i, \mathbf{a}_i)\}_{i=1}^P$. Each $\mathbf{q}_i$ inquires about a knowledge base relation on one of the real-world entities $o_j$, but referring to it via its relation to the fake entity.

to the real-world entity directly, the propagation question will refer to it using its relation to the fake entity $\mathbf{s}_f$ (e.g., *the country where Adam Jacobson was born*). Therefore, the LM must be able to combine its prior knowledge about real-world entities and the injected fake entity $s_f$ to answer the question correctly. The dataset generation process is further described in Appendix E.1.

**Final Dataset** We generate 5K instances of `Controlled RippleEdit` and randomly split these into 4K for training the hypernetwork, 500 for validation, and 500 for testing. To evaluate out-of-domain (OOD) generalization, we generate three additional test sets. We generate 350 instances where their real-world entities $(o_i)$ do not appear in the training dataset (but knowledge base relations occur in the training dataset), calling this set OOD (Entity). Analogously, we generate an OOD (Relation) dataset. Lastly, we generate an OOD (Both) dataset, consisting of 350 instances where neither real-world entities nor the knowledge base relations appear in the training dataset.

## 5.1 Experimental Setup

**Model** We further train the model in Section 4.2 with 500 QA pairs involving real-world entities and relations in `Controlled RippleEdit` to make the propagation easier by reinforcing the model's knowledge of the propagation relations. We call this model `Llama-3.2-1B-base-QA`, and this model is used for all comparison methods in this section.

**Metric** We use LLM-as-a-Judge (with GPT-4o-mini) to evaluate the correctness of the predicted answer against the reference answer, as in the prior section. For efficacy measure, we use the model's performance on multi-hop questions, e.g., *"Q: What is the currency of [the country that Adam Jacobson was born in]? A: U.S. Dollar"*. To measure specificity, we evaluate whether the model retains its ability to answer simplified versions of our questions that do not require any updated knowledge, e.g., *"What is the currency of the United States?"*. See examples in Figure 3. Ideally, updates to the model should not degrade its ability to answer these questions.

**Comparison methods** We use the same set of comparison methods described in Section 4.2. For fair comparison, we modify MEMIT and MEND. As they require the fact **f** to be in an input-output format $(\mathbf{x}, \mathbf{y})$, we map **f** into three atomic facts (e.g., *(Adam Jacobson was born in, the U.S.)*); and conduct multi-edit to inject those facts. See examples in Table 7 and details in Appendix E.3. Lastly, we add one more baseline, described below.

**Meta-Aug CPT** This CPT method explicitly uses how the meta-training set was constructed. We augment an injected fact using the gold in-domain relations for that fact, essentially reproducing what would be the propagation questions in the meta-training set, paired with gold answers. For instance, for the example in Figure 1, we would generate *"What is the currency of the country that Adam Jacobson was born? US Dollar"*, *"What is the currency of the country that Adam Jacobson lived in after retirement? Chinese yuan"*, and *"What is the currency of the country that Adam Jacobson spent adult life? South Korean won"*. We denote this method as Meta-Aug CPT. For In-Domain and OOD (Entity) test sets, note that the test queries are included directly in the CPT data. Because the data is dynamically generated for each example, In-domain and OOD (Entity) are distributionally equivalent for this approach.

Table 2: **Single-edit results on `Controlled RippleEdit`** with `Llama-3.2-1B-base-QA`. We use the model's LLM-Score on multi-hop questions for efficacy, and the model's LLM-Score on single-hop questions for specificity. MEND+Propagation significantly outperforms most of the baselines on the In-Domain test set and also OOD test sets where components of the injected facts do not appear during meta-training. $^\dagger$means the system is out-performed by MEND+Propagation according to a paired bootstrap test ($p = 0.05$). Gray cells do not match the data or learning condition of other cells; see text. For all CPT methods, we conduct full finetuning.

| | #token | In-Domain (2284) | | OOD (Entity) (1368) | | OOD (Rel) (421) | | OOD (Both) (447) | |
|---|---|---|---|---|---|---|---|---|---|
| | | Effi. | Spec. | Effi. | Spec. | Effi. | Spec. | Effi. | Spec. |
| `Llama-3.2-1B-base-QA` | 0× | 8.3$^\dagger$ | 94.7$^\dagger$ | 7.1$^\dagger$ | 94.3 | 8.9$^\dagger$ | 94.2 | 10.9$^\dagger$ | **90.7** |
| Prepend | 1× | 38.1$^\dagger$ | 86.2$^\dagger$ | 41.5 | 88.2 | 29.4$^\dagger$ | 82.4 | 31.7 | 79.5 |
| CPT | 1× | 18.1$^\dagger$ | 80.2$^\dagger$ | 17.0$^\dagger$ | 79.9$^\dagger$ | 15.6$^\dagger$ | 79.3$^\dagger$ | 12.9$^\dagger$ | 71.1$^\dagger$ |
| Meta-Aug CPT | 7× | **80.3** | 75.4$^\dagger$ | 79.1 | 73.3 | 26.1$^\dagger$ | 57$^\dagger$ | 12.9$^\dagger$ | 51.7$^\dagger$ |
| Active-Reading CPT | 95× | 19.6$^\dagger$ | 69.6$^\dagger$ | 19.1$^\dagger$ | 67.8$^\dagger$ | 23.3$^\dagger$ | 65.4 | 16.7 | 62.8 |
| **MEMIT** (`wikitext-103`) | 1× | 12.8$^\dagger$ | 94.4$^\dagger$ | 14.4$^\dagger$ | 94.4 | 12.0$^\dagger$ | 93.9 | 13.8$^\dagger$ | 90.0 |
| **MEMIT** (`Ctrl RippleEdit`) | 1× | 12.0$^\dagger$ | 94.6$^\dagger$ | 13.3$^\dagger$ | **94.5** | 11.1$^\dagger$ | **94.3** | 11.6$^\dagger$ | 90.2 |
| **MEND** (with standard config) | 1× | 14.7$^\dagger$ | 89.0$^\dagger$ | 14.2$^\dagger$ | 89.4 | 10.1$^\dagger$ | 91.8 | 10.7$^\dagger$ | 86.3 |
| **MEND** (Mid-Upper) | 1× | 12.3$^\dagger$ | 91.8$^\dagger$ | 11.5$^\dagger$ | 92.9 | 11.5$^\dagger$ | 92.2 | 12.0$^\dagger$ | 88.1 |
| **MEND+Propagation** (Mid-Upper) | 1× | 60.8$^\dagger$ | 91.3$^\dagger$ | **36.0** | 85.4 | 28.4$^\dagger$ | 87.4 | **18.3** | 84.0 |
| **MEND+Propagation** | 1× | 76.7 | **95.5** | 35.2 | 81.6 | **34.5** | 84.0 | **18.3** | 77.5 |

## 5.2 RESULTS: EFFECTIVENESS OF PROPAGATION

We report the results on `Controlled RippleEdit` in Table 2. Our variant, MEND+Propagation, substantially outperforms other parametric methods consistently for various settings. On the in-domain test set, MEND+Propagation even outperforms Prepend, showing that parametric propagation can be as effective as in-context augmentation.

We observe MEND+Propagation's performance degrades in out-of-domain settings when either entities or relations are unobserved during training. However, MEND+Propagation still outperforms other methods substantially. For example, on OOD (Entity), the best-performing baseline MEMIT (`wikitext-103`) achieves 20.8% lower performance than our variant. We observe that our variant's performance improvement in OOD (Entity) tends to be higher than OOD (Relation). On OOD (Both), where our variant does not observe any entity or relation in the test, MEND+Propagation is able to offer better propagation than others, but the gap is smaller.

**Additional Results** In Appendix F, we report further results on `Controlled RippleEdit`, which we summarize here. In Table 14 and 19, we conduct ablation studies and show design elements proposed in Section 3 are essential. In Table 11 and 18, we conducted experiments that scale hypernetwork in terms of meta-training data and parameter usage, and show OOD performances are not fixable by mere scaling and require some algorithmic innovation. In Table 15, we report performance on two evaluation subsets (where the answer occurred in the meta-training set or not), showing that the hypernetwork generalizes to queries whose answers are not in the meta-training set. In Table 28 and 21, we show that gains from MEND+Propagation are robust to models of different size and family. Lastly, we show MEND+Propagation is cost-effective by comparing memory usage and runtime in Table 20 and 17.

## 6 PROPAGATING MULTIPLE FACTS AT ONCE

Knowledge propagation can be thought as making multiple knowledge edits at once, including injected facts and facts that are impacted by the injected facts. Being able to propagate multiple facts to the language model will have an additional multiplier effect, thus making the learning of knowledge very data-efficient and the updated language model more generalizable. In this section, we extend our approach to RLEdit (Li et al., 2025), a recent hypernetwork training recipe for sequentially injecting multiple facts.

**Setting** We investigate injecting a total of 10 or 20 facts into the language model and measure the propagation performance. With RLEdit, both settings, we inject 5 facts at each turn, and change

Table 3: Multi-edit results on `Controlled RippleEdit` with `Llama-3.2-1B-base-QA`. Training MEND+Propagation with 20 edits leads to out-of-memory in our hardware configuration. Gray cells do not match the data condition of other cells; see text. For all CPT baselines, we conduct full finetuning. We bold the **best** system and underline the second best system.

| LLM-Score (↑) | #token | In-Domain (2284) | | OOD (Entity) (1368) | | OOD (Rel) (421) | | OOD (Both) (447) | |
|---|---|---|---|---|---|---|---|---|---|
| | | Effi. | Spec. | Effi. | Spec. | Effi. | Spec. | Effi. | Spec. |
| `Llama-3.2-1B-base-QA` | 0× | 8.3 | 94.7 | 7.1 | 94.3 | 8.9 | 94.2 | 10.9 | 90.7 |
| *# injected fact = 10* | | | | | | | | | |
| CPT | 1× | 11.2 | 88.6 | 9.3 | **85.9** | 14.5 | **89.0** | 12.8 | **80.7** |
| Meta-Aug CPT | 7× | **77.3** | 88.6 | 76.8 | 86.6 | **31.8** | 73.6 | **17.9** | 63.0 |
| Active-Reading CPT | 95× | 12.8 | 75.9 | 10.3 | 77.0 | 17.3 | 78.0 | 14.7 | 74.0 |
| MEND+Propagation | 1× | 25.5 | 68.2 | 15.5 | 48.4 | 10.9 | 54.6 | 11.4 | 56.5 |
| RLEdit+Propagation | 1× | 48.6 | **92.9** | **25.8** | 82.0 | 8.6 | 87.8 | 17.6 | 79.5 |
| *# injected fact = 20* | | | | | | | | | |
| CPT | 1× | 10.3 | **89.8** | 8.3 | **88.2** | 13.8 | **89.9** | 12.5 | **82.5** |
| Meta-Aug CPT | 7× | **72.2** | 89.2 | 68.8 | 85.9 | **28.3** | 74.5 | **17.0** | 62.7 |
| Active-Reading CPT | 95× | 10.5 | 77.0 | 9.1 | 76.2 | 17.1 | 77.3 | 16.4 | 76.8 |
| RLEdit+Propagation | 1× | 29.9 | 88.5 | **18.3** | 78.3 | 12.0 | 82.6 | 13.0 | **82.5** |

the number of turns to control the total number of facts. Since MEMIT and MEND underperform CPT-based methods on the single-edit scenario, and prior work (Meng et al., 2023; Mitchell et al., 2022) shows that multi-edit performance mostly decreases from single-edit performance, we focus on CPT-based baselines in this section.

**Results** Table 3 reports performances. All methods show degraded performances compared to single edit scenarios, especially as the number of edits grows. Meta-Aug CPT shows strongest efficacy overall, especially in-domain. RLEdit+Propagation outperforms MEND+Propagation in most settings by a large margin. RLEdit+Propagation shows competitive performance in OOD (Entity), and comparable performance in OOD (Both), but lags in OOD (Relation), compared to CPT methods. Active-Reading CPT and Meta-Aug CPT, while showing stronger efficacy performance in OOD (Both) setting, also show a specificity drop.

## 7 RELATED WORK

**Knowledge Propagation** Recent work has studied the propagation of injected knowledge, finding that existing methods are largely lacking. A line of work (Ma et al., 2024; Berglund et al., 2024) studied reversal curse — the model knows "A is B", but not "B is A". Other work (Qin et al., 2024; Nishi et al., 2025) analyzes unintended ripple effects of different editing methods. Hase et al. (2024) surveys a wide range of open problems regarding revising the belief of the model. We discuss recent benchmarks for evaluating knowledge edits in Appendix H.1.

**Continual Learning** Knowledge editing can be viewed as continual learning, injecting new knowledge gradually. Continual learning has been studied in domain adaptation scenarios (Gururangan et al., 2020; Ke et al., 2023). A line of work studies catastrophic forgetting during continual learning (Chen et al., 2025; Franke et al., 2024; Jin & Ren, 2024a;b). They evaluate the performance on downstream tasks, rather than changes in parametric knowledge. We include more discussion related to continued pretraining in Appendix H.2.

## 8 CONCLUSION

In this work, we introduce a simple but effective modification that addresses the critical challenge of propagating edits to related facts in current knowledge editing techniques. We show the effectiveness of our method on `RippleEdit`, a widely-adopted dataset measuring propagation. We present a controlled dataset centered around well-known entities and relations to further demonstrate the effectiveness when propagated knowledge is known by the model. We also show that our modification is generalizable to multi-edit editing methods, and the modified variant maintains strong performance improvement over baselines.

**Reproducibility Statement** Our code and dataset will be made publicly available once. See anonymized code at `https://anonymous.4open.science/r/propmend-84F6/README.md`. We provided the description of `RippleEdit` at Appendix D; and `Controlled RippleEdit` at Section 5 and Appendix E.

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

# APPENDIX

## A PROMPT

### A.1 LLM-as-Judge prompt

```
[Instruction]
Please act as an impartial judge and evaluate the quality of
the response provided by an AI assistant to the user question
displayed below.  For this evaluation, you should primarily
consider the following criteria:
accuracy:
Score 0:  The answer is completely unrelated to the reference.
Score 3:  The answer has minor relevance but does not align with
the reference.
Score 5:  The answer has moderate relevance but contains
inaccuracies.
Score 7:  The answer aligns with the reference but has minor
omissions.
Score 10:  The answer is completely accurate and aligns
perfectly with the reference.
Only respond with a numerical score.

[Question]
{question}

[The Start of Ground truth]
{reference}
[The End of Ground truth]

[The Start of Assistant's Answer]
{prediction}
[The End of Assistant's Answer]

Return the numerical score wrapped in <score>..</score> tag
```

# B DETAILS ON BASELINE METHODS

## B.1 PREPEND

We follow the practice in (Cohen et al., 2024) and format the prepended text to be "`Imagine that` **f**", where **f** is the injected fact.

## B.2 ACTIVE-READING CPT

Active-Reading generates augmentation in two stages: 1) generate task-agnostic and task-specific learning strategies for each fact; 2) given a generated strategy and the fact, generate the corresponding augmentation for learning the fact. We use `gpt-5-nano` to generate the augmentations. We note that this approach reported stronger performance than a similar augmentation approach, EntiGraph (Yang et al., 2024).

## B.3 MEMIT

MEMIT (Meng et al., 2023) frames knowledge editing as an optimization problem to compute the updated weights. This method assumes three inputs: the verbalization of `subject-relation` **x**, the string corresponding to `subject` $s$, and the string corresponding to `object` $o^*$. For the optimization to run effectively, the approach precomputes a covariance matrix (per target weight) from a reference corpus, typically, `wikitext-103` (Merity et al., 2017). To reconcile potential train-test mismatch, we precompute the covariance matrix on the meta-training set of MEND+Propagation, using both the injected facts, and the propagation query-answer pairs. See hyperparameters used in Appendix G.

## B.4 MEND

Our work follows the same hypernetwork structure as MEND (Mitchell et al., 2022). We describe their design choices here, which are also adopted by our approach. Their algorithm is shown in Figure 4.

**MEND on `RippleEdit`**  MEND (with standard config) is trained on the zsRE question-answering dataset (Levy et al., 2017) with their original hyperparameters (editing top 3 MLP layers (i.e., Layer-`[13-15]`)). Similar to our practice in MEMIT, we also use the meta-training set that MEND+Propagation uses; and we target at Mid-Upper Layers editing Layer-`[10-12]` according to our hyperparameters investigation at Appendix C. This provides most controlled comparison setting with our method, we also target at (denoted MEND (Mid-Upper)). We use `gpt-4o` to create a paraphrased input $\mathbf{x}'$ required for training.

At test time, MEND uses Supervised Fine-Tuning loss to create the gradient input to the hypernetwork, with a verbalized prefix of subject-relation $(s, r, \cdot)$ as input and new object $o^*$ as output. To train the hypernet, one need paraphrase of $(s, r, \cdot)$.

**Rank-1 matrix decomposition**  Consider a specific weight matrix $W \in \mathcal{W}$. Let $\delta \in \mathbb{R}^m$ be the gradient of the loss with respect to the output of $W$; and $u \in \mathbb{R}^d$ be the input to the weight $W$. MEND observes that the gradient of the loss with respect to $W$, $\nabla_{\mathcal{W}} L^I$, is decomposable by the outer product between $\delta$ and $u$, namely $\delta u^\top$. The calculation can be extended to a batch instances via $\sum_{i=1}^B \delta^i u^{i^\top}$, where superscipt $i$ denotes corresponding values for instance $i$. Due to this observation the hypernetwork $g_\phi$ parameterized by $\phi$ could operate on $\delta^i$ and $u^i$ as input without loss of information; correspondingly, it could output values $\tilde{u}$ and $\tilde{\delta}$ to compose the proposed update gradient through outer product $\tilde{\nabla}_W = \tilde{\delta} \tilde{u}^\top$. Finally, we compute $W \leftarrow W - \alpha \tilde{\nabla}_W$, where $\alpha$ is a learned weight-specific step size. This observation drastically reduces the computation cost of hypernetwork from $O(d \times m)$ to $O(d + m)$ and make training the hypernetwork feasible.

**Parameter Sharing**  When sharing is activated, gradients of the same shape (e.g., MLP down-projection in layer 10 and layer 12) will be modified by the same hypernetwork. To enable some layer-wise specialization, MEND applies a layer-specific scale and offset to the editor network

Figure 4: MEND algorithm; reproduced from Mitchell et al. (2022)

| **Algorithm 1** MEND Training (Outer Loop) | **Algorithm 2** MEND Edit Procedure (Inner Loop) |
|---|---|
| 1: **Input:** Pre-trained $p_\theta$, weights to make editable $\mathcal{W} \subseteq \theta$, editor params $\phi$, edit dataset $D^{tr}_{edit}$, edit-locality tradeoff $c_{\text{edit}}$ | 1: **procedure** EDIT$(\theta, \mathcal{W}, \phi, \mathbf{x}, \mathbf{y})$ |
| 2: **for** $t \leftarrow 1, 2, ...$ **do** | 2: $\quad \hat{p} \leftarrow p_\theta(\mathbf{y} \mid \mathbf{x})$, **caching** input $u_\ell$ to $W_\ell \in \mathcal{W}$ |
| 3: $\quad$ Sample $\mathbf{x}, \mathbf{y}, \mathbf{x}', \mathbf{x}_{\text{loc}} \sim D^{tr}_{edit}$ | 3: $\quad L^I(\mathbf{x}, \mathbf{y}) \leftarrow -\log \hat{p} \qquad \triangleright$ Compute neg log-likelihood |
| 4: $\quad \tilde{\mathcal{W}} \leftarrow \text{EDIT}(\theta, \mathcal{W}, \phi_{t-1}, \mathbf{x}, \mathbf{y})$ | 4: $\quad$ **for** $W_\ell \in \mathcal{W}$ **do** |
| 5: $\quad L_{\text{e}} \leftarrow -\log p_{\tilde{\mathcal{W}}}(\mathbf{y} \mid \mathbf{x}')$ | 5: $\quad\quad \delta_{\ell+1} \leftarrow \nabla_{W_\ell u_\ell} L^I(\mathbf{x}, \mathbf{y}) \qquad \triangleright$ Grad w.r.t. output |
| 6: $\quad L_{\text{loc}} \leftarrow \text{KL}(p_{\mathcal{W}}(\cdot \mid \mathbf{x}_{\text{loc}}) \| p_{\tilde{\mathcal{W}}}(\cdot \mid \mathbf{x}_{\text{loc}}))$ | 6: $\quad\quad \tilde{u}_\ell, \tilde{\delta}_{\ell+1} \leftarrow g_{\phi_\ell}(u_\ell, \delta_{\ell+1}) \qquad \triangleright$ Rank-1 udpate vec |
| 7: $\quad L^O(\phi_{t-1}) \leftarrow c_{\text{edit}} L_{\text{e}} + L_{\text{loc}}$ | 7: $\quad\quad \tilde{\nabla}_{W_\ell} \leftarrow \tilde{\delta}_{\ell+1} \tilde{u}_\ell^\top \qquad \triangleright$ Compose the full update grad |
| 8: $\quad \phi_t \leftarrow \text{Adam}(\phi_{t-1}, \nabla_\phi L(\phi_{t-1}))$ | 8: $\quad\quad \tilde{W}_\ell \leftarrow W_\ell - \alpha_\ell \tilde{\nabla}_{W_\ell} \qquad \triangleright$ Learned step size $\alpha_\ell$ |
| | 9: $\quad \tilde{\mathcal{W}} \leftarrow \{\tilde{W}_1, ..., \tilde{W}_k\};$ **return** $\tilde{\mathcal{W}}$ |

hidden state and output, similar to FiLM layers (Perez et al., 2018). For the set of target weights $\mathcal{W}$, parameter sharing reduces computation costs of training the hypernetwork from $O(|\mathcal{W}| \cdot (d+m))$ to $O(c \cdot (d+m))$ for some constant $c$; in this study, since MLPs only have two distinct weight sizes (i.e., down-projection and up-projection), the constant $c = 2$. The recommended setting from MEND (Mitchell et al., 2022) is to do parameter sharing. We also follow the same setting.

## C   MEND HYPERPARAMETERS INVESTIGATION

We re-investigate the hyperparameters and design choices of MEND, and we find that the choice of layers for parameter updating impacts the model's performance. MEND and other methods, such as MEMIT, selectively target certain layers within the LLM to modify. In MEND, the default configuration is to have the hypernetwork target the MLPs weights of the top 3 layers; however, we find editing lower layers is more effective for knowledge propagation. Applying the hypernetwork to all layers is expensive, since the hypernetwork operations are memory-intensive. Table 24c in the appendix reports the layers modified with MEND+Propagation. We adopt the same choice of layers for RLEdit+Propagation as well.

## D   RIPPLEEDIT

**Task**   In RippleEdit (Cohen et al., 2024), given an original (subject, relation, object) triplet $(s, r, o)$, an edit (e.g., $o \rightarrow o^*$) is constructed to form a new triplet $\mathbf{e} = (s, r, o^*)$. The new triplet can be mapped into a natural language sentence with a template, which we denote as $\mathbf{f}$. Each edit can incur changes in other existing fact triplets.

RippleEdit captures propagation by identifying and preparing tests queries for 6 propagation types: 1. Logical Generalization (LG), a related fact that is created as a logical by-product of the relation $r$ (e.g., brother); 2. Compositionality I (CI), a multi-hop fact composed with another fact about the target object $o^*$; 3. Compositionality II (CII), a multi-hop fact that uses a different subject $s'$ but still holds for the new object $o^*$; 4. Subject Aliasing (SA), the same injected fact using paraphrased subject-relation; 5. Forgetfulness (FN), a neighbor triplet whose answer $o'$ does not change despite sharing the same relation $r$ as the edit (i.e., $r$ is a one-to-many relation); 6. Relation Specificity (RS), another fact about the subject $s$ that's not affected by the edits. See examples in Table 4.

The dataset is released under the MIT License, and is available at https://github.com/edenbiran/RippleEdits/tree/main/data/benchmark.

Table 4 shows examples of various propagation types. The example is adapted from (Cohen et al., 2024). In Table 5, we include a table showing what percentage of propagation questions per propagation type have one of their valid answers in the injected fact.

In Table 6, we include a table showing how many propagation questions are included per propagation type.

Table 4: `RippleEdit` example across various propagation types. The example is adapted from Cohen et al. (2024).

(a) A snapshot of world knowledge at the time of edit.

| Entity | Knowledge Triplets |
|---|---|
| Prince
④ (Prince, `alias`, Prince Roger Nelson) | ① (Prince, `sibling`, Tyka Nelson)
② (Tyka Nelson, `profession`, Singer)
③ (Prince, `founder_of`, Paisley Park Records)
⑤ (Mattie Shaw, `mother_of`, Prince) |
| Nicholas Carminowe | ⑥ (Nicholas Carminowe, `profession`, Members of Parliament)
⑦ (Nicholas Carminowe, `sibling`, John Carminowe) |

(b) Edit that introduce changes among entities.

| New relation created |
|---|
| ⑧ (Prince, `sibling`, Nicholas Carminowe) |

(c) Propagation that follows from the edit in Table 4b. We highlight the use of injected fact ⑧, and the cases where certain knowledge is expected to be **[Not forgotten]**.

| Propagation type | Question | Answer (Explanation) |
|---|---|---|
| Logical Genralization | The siblings of Nicholas Carminowe are | Prince (⑧ + `sibling` is a symmetric relation)
John Carminowe (⑥) |
| Compositionality I | The professions of the siblings of Prince are | Members of Parliament (⑧ + ⑤)
Singer (① + ②) |
| Compositionality II | The siblings of the founder of Paisley Park Records are | Nicholas Carminowe (③ + ⑧)
Tyka Nelson (③ + ①) |
| Subject Aliasing | The siblings of Prince Roger Nelson are | Nicholas Carminowe (④ + ⑧)
Tyka Nelson (④ + ①) |
| Forgetfulness | The siblings of Prince are | Nicholas Carminowe (⑧)
Tyka Nelson (①) **[Not forgotten]** |
| Relation Specificity | The mother of Prince is | Mattie Shaw (⑧) **[Not forgotten]** |

Table 5: Percentage of verbatim question in `RippleEdit`, where the one of the valid answers $a \in \mathcal{A}_i$ appeared in the edit fact in test examples.

| Propagation Query Type | Train set | Validation set | Test set |
|---|---|---|---|
| Percentage of verbatim question in Logical Generalization | 35.8% | 51.8% | 55.2% |
| Percentage of verbatim question in Compositionality I | 11.0 | 12.3% | 11.7% |
| Percentage of verbatim question in Compositionality II | 100.0% | 100.0% | 100% |
| Percentage of verbatim question in Subject Aliasing | 100.0% | 100.0% | 100% |
| Percentage of verbatim question in Relation Specificity | 3.2% | 3.5% | 3.2% |
| Percentage of verbatim question in Forgetfulness | 87.4% | 79.3% | 81.9% |
| Overall | 31.3% | 32.1% | 31.9% |

Table 6: Verbatim rate on test examples. Percentage of `RippleEdit` propagation questions where one of the valid answers $a \in \mathcal{A}_i$ appeared in the edit fact in test examples.

| Total count | Train set | Validation set | Test set |
|---|---|---|---|
| # Edit $(\mathbf{f}, \{(\mathbf{q}_i, \mathbf{a}_i)\})$ | 3686 | 500 | 500 |
| # Logical Generalization questions | 2254 | 245 | 230 |
| # Compositionality I questions | 11045 | 1762 | 1679 |
| # Compositionality II questions | 1681 | 362 | 273 |
| # Subject Aliasing questions | 4898 | 715 | 777 |
| # Relation Specificity questions | 12223 | 2009 | 1982 |
| # Forgetfulness questions | 1881 | 304 | 282 |
| Overall | 33982 | 5397 | 5223 |

Table 7: An example instance of `Controlled RippleEdit`. As mentioned in Section E.3, since some baselines require facts to be in input-output format, we also show an example for the processing.

| | |
|---|---|
| $\mathbf{f}$ | *[Elizabeth Ruiz]*$s_f$ was born in **[Kenya]**$o_1$. She spent most of her adult life in **[Malaysia]**$o_2$. After retirement, she lived in **[Egypt]**$o_3$ and passed away. |
| $\mathbf{q}_i, \mathbf{a}_i$ | What is the capital city of the country that *[Elizabeth Ruiz]*$s_f$ spent most of her adult life in?, Kuala Lumpur |
| $\hat{\mathbf{q}}_i, \mathbf{a}_i$ | What is the capital city of **[Malaysia]**$o_2$?, Kuala Lumpur |
| 3 Atomic facts $(\mathbf{x}, \mathbf{y})$ | ( *[Elizabeth Ruiz]*$s_f$ was born in, **[Kenya]**$o_1$ )
( *[Elizabeth Ruiz]*$s_f$ spent most of her adult life at, **[Malaysia]**$o_2$ )
( *[Elizabeth Ruiz]*$s_f$ died in, **[Egypt]**$o_3$ ) |

# E CONTROLLED RIPPLEEDIT

In this section, we discuss implementation details regarding our controlled synthetic dataset `Controlled RippleEdit`. First, we discuss how we generate the components of our dataset (i.e., the well-known entities and relations) in Section E.1. Then, we describe how we conduct further filtering to a smaller set of entities and relations in Section E.2. We describe how we conduct additional preprocessing for baselines MEND and MEMIT in Section E.3.

## E.1 DATA GENERATION

**Generating the initial list of well-known entities and relations** We prompt ChatGPT to generate a list of head entities per entity type and manually filter out invalid entities. Then, starting from a list of general questions from ChatGPT, we manually iterate to obtain general relations per entity type. In generating the relation per entity type, we specifically aim for a general relation template that could be asked about any kind of entity within that type and could be answered with a short answer. Then, we programmatically generate all single-hop questions by instantiating each template with entity name. We prompt GPT-4.1 for answer or "*I don't know*". After filtering for where answers are provided, we reprompt the model to shorten any answer that's longer than 30 characters. We treat the answer from GPT-4.1 as the gold answer; we observed this to be extremely reliable on instances that we manually inspected due to the well-known nature of the entities and relations.

**Generate facts and questions** Given a list of well-known entities and relations, we follow the following process in all cases to generate fact and its paired questions: (1) sample an entity type, where the probability of sampling an entity type determined by the number of entities of that type and whether that type has at least 1 relation; (2) randomly choose 3 entities from the list of entities of that type; (3) randomly choose which entity (among the 3 entities) to construct the efficacy and specificity question, for each relation of that entity type; (4) apply templates to arrive at facts and questions.

**Dataset Generation** We manually select seven high-level categories for real-world entities: person, event, language, creative work, organization, species, and country. We manually design two fact templates per entity type, where one story template assumes the fake entity to be a person and the other a company. Figure 3 shows an example where the type of the fake entity is person and the type of the real-world entity is country.

For each entity type, we prompt an LLM to generate (1) a list of entities belonging to the entity type and (2) relations relevant to the entity type. To effectively test propagation, we aim to restrict the entities and relations to those that are largely "known" by LLMs. Therefore, we filter datasets to obtain a smaller set of real-world entities (a total of 189 unique entities) and relations (a total of 38 unique relations). See more description in Appendix E.2.

From this set, we randomly sample three real-world entities of the same type and use fact template to generate fact to be injected. We can now form efficacy questions, querying relations on the real-world entities in the fact.

## E.2 DATASET FILTERING

We initially start with a set of 760 real-world entities and 48 relations. We filter this set to remove entities and relations not well-known to base LLMs. Specifically, we start with `Llama-3.2-1B-base-QA`) model. For each of 48 relations, we sample 10 real world entities and further train `Llama-3.2-1B-base-QA`) model with those 480 examples.

With this model, we query all valid real-world entity, relation pairs. We use LLM-as-a-Judge to compare the predicted answer and GPT-4.1 answer, providing a score between 0 and 1. Then, we only keep pairs with LLM-as-a-Judge score higher than 0.4. For each entity type, all entities belonging to it have the same number of relations, the number of entities is at least 20, and the number of relation is at least 4. **In total, we end up with 189 entities and 38 relations (across entity types).** See the full list of entities in Table 9; see the list of relations in Table 10 and the list of entities in Table 9.

### E.3 BASELINES

**Prepend**  We mildly modify the prompt from (Cohen et al., 2024) to maintain grammaticality: for fake person as the subject, we use "`Imagine that someone named`  **f**"; and for fake company as the subject, we use "`Imagine that a company named`  **f**".

**Modifications for MEMIT and MEND**  MEMIT and MEND require the fact to be in an input-output format $(\mathbf{x}, \mathbf{y})$ and uses Supervised Fine-Tuning (SFT) loss $-\log p(\mathbf{y} \mid \mathbf{x})$, where output $\mathbf{y}$ is the real-world object $o_r$. For MEMIT, the input $\mathbf{x}$ is a verbalization for fake entity $s_f$ and the relation being tested $r$; and the name of the fake entity must be a substring of the verbalization. Although MEND does not require access to a substring of fake entity $s_f$, it requires a paraphrase of input $\mathbf{x}'$ for meta-training. Because story and question are template-generated, we also curate the templates to generate those components for each story template.

## F  CONTROLLED RIPPLEEDIT ADDITIONAL RESULTS

**Scaling up**  We increase the hypernetwork size and the amount of meta-training data in Table 11 to investigate whether further scaling of the hypernetwork can lead to stronger performance. We find that increasing both can lead to substantial performance gains. However, although in-domain performance is close to perfect after scaling up both factors, increasing OOD performance remains a challenge.

**Ablation of MEND+Propagation Design Choices**  Table 14 and 19 present ablations of the MEND+Propagation design choices. First, we investigate having paraphrased inputs in the outer loop of MEND+Propagation (Mid-Upper), similar to MEND, instead of propagation questions in the outer loop. This change is the most impactful one; without it, we see substantial performance degradation, suggesting that the hypernetwork training needs to be aligned with its intended test scenario. Second, we investigate changing the loss in the inner loop. In MEND+Propagation (Mid-Upper), we apply the causal language modeling on all tokens of the fact **f**. To change to SFT, we map the fact **f** into three atomic facts taking an input-output format $(\mathbf{x}, \mathbf{y})$ (e.g., *(Adam Jacobson was born in, the U.S.)*, see full example in Table 7); and the loss is calculated on the answer tokens $\mathbf{y}$ given the input $\mathbf{x}$. Training on all tokens as we do in MEND+Propagation (Mid-Upper) works substantially better in-domain, but in some OOD settings training on just answer tokens is competitive. Finally, we also find it is more effective to edit the Mid-Upper layers than the Upper layers of the transformer. In Table 19, we show an ablation study with MEND+Propagation (Mid-Upper), and observe similar finding as in Table 14.

**Stratified analysis**  We investigate whether the trained hypernetwork is merely memorizing answers in meta-training set or whether it also generalize. We further split propagation by the criteria of whether the gold answer appears verbatim in meta-training set, and we repeat the operation for our three out-of-domain test sets. Table 15 shows that MEND+Propagation substantially improves over baselines on OOD (Entity) for both types of propagation questions. However, the performance gap on questions whose answers are not in meta-training set reduces noticeably. We believe that this shows our method is able to generalize but have rooms for future improvements.

**Runtime Efficiency Evaluation**  We report the efficiency of various editing methods, measured by their max memory usage and total runtime in Table 17 and 20. "Base Model" does not involve any editing and only incurs inference costs. Different editing methods show different trade-offs between memory usage and runtime, and CPT (Full) is the least efficient in both dimensions. MEND+Propagation (Mid-Upper) is similarly efficient to MEND when editing the same number of layers, and gets less efficient when editing more layers. In Table 20, although both MEND+Propagation (Mid-Upper) and MEND+Propagation's parameter counts is 12.8% of `Llama3.2-1B-base-QA`, the number of layers being edited is the dominant factor in memory and runtime. Similar pattern is observed in Table 17.

**Results with Other Base Models**  We report experimental results with `Llama3.2-1B-base-QA` and `Llama3.2-3B-base-QA` in Table 2 and Table 28 in the appendix. We observe very similar experimental trends when editing `Qwen-2.5-1.5B-base-QA`, showing that the results from

Table 8: Story templates of all entity types.

| Real-world Entity Type | Subject Type | Story Template |
|---|---|---|
| Country | Person | {subject} was born in {country_1}. {Gender_subj} spent most of {gender_possessive_adj} adult life in {country_2}. After retirement, {gender_subj} lived in {country_3} and passed away. |
| | Company | {subject} was founded in {country_1}. {Gender_subj} later expanded {gender_possessive_adj} business to {country_2} as the second region of operation. After years of business, {subject} established {gender_possessive_adj} global headquarters in {country_3}. |
| Person | Person | {subject} first wrote about {person_1} in an 8th-grade book report. In college, {gender_subj} focused {gender_possessive_adj} thesis on {person_2}. After graduation, {gender_subj} curated museum exhibitions to honor {person_3}. |
| | Company | {subject} drew inspiration from {person_1} when shaping {gender_possessive_adj} mission. Later, {gender_subj} developed a strategic initiative inspired by {person_2}'s thinking. Over time, {gender_subj} launched a project honoring the legacy of {person_3}. |
| Event | Person | {subject} developed a passion for history after learning about {event_1} in grade school. In college, {gender_subj} did research on {event_2}. Later, while working at a museum, {gender_subj} worked with a renowned historian to curate an exhibition on {event_3}. |
| | Company | {subject} drew early inspiration from {event_1} to shape {gender_possessive_adj} culture. Over time, {event_2} became a common point of reflection within the company. Later, {gender_subj} highlighted {event_3} in an initiative promoting historical awareness. |
| Species | Person | {subject} became fascinated with nature after learning about {species_1}. During graduate school, {gender_subj} researched on {species_2}. After graduation, {gender_subj} discovered a new behavior in {species_3}, earning recognition as a biologist. |
| | Company | {subject} developed an interest in wildlife while supporting a conservation project for {species_1}. {Gender_subj} later partnered with researchers to study {species_2}. {Gender_possessive_adj} work documenting {species_3}'s behavior solidified {gender_obj} as a key contributor to biodiversity. |
| Language | Person | {subject} was born into a {language_1}-speaking environment. In grade school, {gender_subj} started to learn {language_2}. In {gender_possessive_adj} college, {gender_subj} took a major in {language_3}. |
| | Company | {subject} began by offering services in {language_1}. {Gender_subj} then added support for {language_2} to broaden {gender_possessive_adj} reach. Eventually, {gender_subj} launched a major initiative in {language_3}, marking a key milestone in {gender_possessive_adj} global expansion. |
| Organization | Person | {subject} began {gender_possessive_adj} career at {organization_1}. After years of hard work, {gender_subj} became a manager at {organization_2}. Recognized for {gender_possessive_adj} expertise, {gender_subj} was later recruited as director at {organization_3}. |
| | Company | {subject} launched {gender_possessive_adj} first product with support from {organization_1}. {Gender_subj} later collaborated on a major project with {organization_2}. Eventually, {subject} was acquired by {organization_3}. |
| Creative Work | Person | {subject} discovered a passion for creative work after encountering {creative_work_1}. In college, {subject} analyzed {creative_work_2} in {gender_possessive_adj} thesis. Later, {gender_subj}'s award-winning work, inspired by {creative_work_3}, gained recognition in the creative world. |
| | Company | {subject} built {gender_possessive_adj} culture on the influence of {creative_work_1}. Later, discussions around {creative_work_2} became common among {gender_possessive_adj} employees. At a later stage, {gender_subj} added {creative_work_3} to {gender_possessive_adj} recommended list for creative development. |

Table 9: All real-world entities in `Controlled RippleEdit`.

| In-Domain / Out-of-Domain | Real-world Entity Type | Entity Instances |
|---|---|---|
| In-Domain | Person | Martin Luther King Jr., Napoleon Bonaparte, William Wordsworth, William Shakespeare, Genghis Khan, Vincent van Gogh, Mother Teresa, Leonardo da Vinci, Eleanor Roosevelt, Theodore Roosevelt, Albert Einstein, Cleopatra VII, Frida Kahlo, Pablo Picasso, Rosa Parks, Elvis Presley, Joan of Arc, Franklin D. Roosevelt, Marie Antoinette, Henry VIII, Coco Chanel |
| | Language | Polish, Portuguese, English, Hindi, Swedish, German, Spanish, Turkish, Greek, Persian (Farsi), Hebrew, French, Arabic, Gujarati, Bengali, Dutch, Korean, Tamil, Telugu, Italian, Kazakh, Haitian Creole, Punjabi, Swahili |
| | Country | Iran, Malaysia, Colombia, Kenya, Armenia, Israel, Maldives, Vietnam, Saudi Arabia, Pakistan, Bangladesh, Turkey, Germany, Czech Republic, United States, Russia, Ukraine, Oman, Japan, South Korea, Belgium, Norway, New Zealand, Indonesia, Denmark, France, India, Spain, Iceland, Greece, Thailand |
| | Event | The Reign of Alexander the Great, The Fall of the Berlin Wall, The Spanish Conquest of the Aztecs, The Assassination of Julius Caesar, The Collapse of the Soviet Union, The Battle of Midway, The Surrender of Japan in WWII, Abolition of Slavery in the US, The Establishment of the Ming Dynasty, The Emancipation Proclamation, The Execution of King Louis XVI, The Partition of India and Pakistan, The Assassination of John F. Kennedy, Signing of the Magna Carta, American Civil War, Moon Landing, The Battle of Thermopylae, The Establishment of the People's Republic of China, Fall of Constantinople, The Founding of the United States of America, The Taiping Rebellion, The Vietnam War, The Battle of Waterloo, Civil Rights Movement |
| | Organization | Toyota, Human Rights Watch, Sony, Spotify, The Salvation Army, Amazon, Bill & Melinda Gates Foundation, Apple, The ACLU, Ford, World Food Programme, Amnesty International, Siemens, Johnson & Johnson, World Health Organization, Nestlé, Alibaba, Airbnb, Walmart
What primary service or product does {organization} provide? |
| | Species | pygmy hippo, panda, praying mantis, red-shouldered hawk, swan, humpback whale, crocodile, snow leopard, tiger, king cobra, great horned owl, great white shark, wolverine, bengal tiger, whale shark, bald eagle, wildebeest, harpy eagle |
| | Creative Work | The Brothers Karamazov, Oldboy, The Count of Monte Cristo, Jane Eyre, Citizen Kane, The Hobbit, Gangnam Style, A Tale of Two Cities, War and Peace, Goodfellas, The Dark Knight, Brave New World, Catch-22, Pulp Fiction, The Grapes of Wrath |
| Out-of-Domain | Person | Alexander the Great, Machiavelli, Charles Dickens |
| | Language | Afrikaans, Sinhala, Russian, Malay, Ukrainian |
| | Country | Portugal, Italy, Sweden, Netherlands, Poland, Azerbaijan, Hungary |
| | Event | The Boston Tea Party, The Montgomery Bus Boycott, Protestant Reformation, The Haitian Revolution, Napoleonic Wars, French Revolution, The 9/11 Attacks, English Civil War, The Battle of Hastings |
| | Organization | Walt Disney Company |
| | Species | albatross, raccoon, mantis shrimp, giant panda, giraffe, sloth, chameleon |
| | Creative Work | Pride and Prejudice, The Road, A Separation, Spirited Away, Pan's Labyrinth |

Table 10: All relations in `Controlled RippleEdit`.

| In-Domain / Out-of-Domain | Real-world Entity Type | Relation Template |
|---|---|---|
| In-Domain | Person | What occupation is {person} most well-known for? |
| | | Where was the birthplace of {person}? |
| | | What language was primarily spoken by {person}? |
| | | What year did {person} pass away? |
| | | What is the religion of {person}? |
| | | What year was {person} born? |
| | Language | What writing system is used by {language}? |
| | | What is the ISO 639-1 code for {language}? |
| | | What region is {language} native to? |
| | Country | What is the top-level internet domain for {country}? |
| | | What is the currency of {country}? |
| | | What is the ISO alpha-2 code for {country}? |
| | | Which ethnic group is the largest in {country}? |
| | | What is the capital of {country}? |
| | | What language in {country} has the most speakers? |
| | | What is the calling code for {country}? |
| | Event | In which country did {event} happen? |
| | | Who was the most important leader or figure involved in {event}? |
| | Organization | Where was {organization} established? |
| | | In what year was {organization} established? |
| | | Who established {organization}? |
| | | What is the primary field or industry of {organization}? |
| | | What primary service or product does {organization} provide? |
| | Species | What is the social structure of {species}? |
| | | What is the diet of {species}? |
| | | What type of organism is {species}? |
| | Creative Work | What is the original language of {creative_work}? |
| | | When was {creative_work} released or published? |
| | | Where was {creative_work} produced or created? |
| | | In which country was {creative_work} first released or published? |
| | | What is the genre or style of {creative_work}? |
| Out-of-Domain | Person | ∅ |
| | Language | What is the name of the alphabet or script of {language}? |
| | Country | Which religion has the most followers in {country}? |
| | Event | When did {event} take place? |
| | | What year did {event} end? |
| | Organization | Where is the headquarters of {organization} located? |
| | Species | Where is {species} primarily native to? |
| | Creative Work | Who is the creator of {creative_work}? |

Table 11: Scaled-up experiment of MEND+Propagation on `Controlled RippleEdit` with `Qwen-2.5-1.5B-base-QA`. We experiment with more in-domain meta-training instances, and different sizes of hypernetwork by having dedicated hypernetworks per target weight in `Qwen-2.5-1.5B-base-QA`. We observed that having larger training data and hypernetwork tends to improve performances on Out-of-Domain instances, but it remains challenging.

| LLM-Score (↑) | # Hypernet Param. | # train instances | In-Domain (2284) | | OOD (Entity) (1368) | | OOD (Relation) (421) | | OOD (Both) (447) | |
|---|---|---|---|---|---|---|---|---|---|---|
| | | | Effi. | Spec. | Effi. | Spec. | Effi. | Spec. | Effi. | Spec. |
| MEND+Propagation | 163M | 4K | 64.0 | 93.6 | 34.7 | 83.0 | 33.3 | 84.8 | 17.7 | **85.8** |
| | 3.4B | 30K | **98.5** | **96.0** | **42.2** | **88.6** | **42.9** | **87.4** | **17.8** | 84.0 |

Table 12: Scale-up experiment of MEND+Propagation on `Controlled RippleEdit` with `Llama-3.2-1B-base-QA`. We experiment with more in-domain meta-training instances, and different sizes of hypernetwork by having dedicated hypernetworks per target weight in `Llama-3.2-1B-base-QA`. We observed that having larger training data and hypernetwork tends to improve performances on Out-of-Domain instances, but it remains challenging.

| LLM-Score (↑) | # Hypernet Param. | # train instances | In-Domain (2284) | | OOD (Entity) (1368) | | OOD (Relation) (421) | | OOD (Both) (447) | |
|---|---|---|---|---|---|---|---|---|---|---|
| | | | Effi. | Spec. | Effi. | Spec. | Effi. | Spec. | Effi. | Spec. |
| MEND+Propagation | 159M | 4K | 76.7 | 95.5 | 35.2 | 81.6 | 34.5 | 84.0 | 18.3 | 77.5 |
| | 2.8B | 30K | **97.8** | **97.1** | **42.5** | **87.2** | **41.8** | **89.5** | **20.9** | **87.8** |

MEND+Propagation hold for a different model family and size. We also conducted more extensive experiment with `Llama3.2-1B-base-QA`. See details in Appendix F.

In Table 2, we include full test results with `Llama-3.2-1B-base-QA`. On the in-domain test set, MEND+Propagation outperforms Prepend (the next best performing system) by 35.3%. We also observe performance degradation in out-of-domain settings. When either entities or relations are unobserved during training, MEND+Propagation maintains a strong performance gap with other methods. For example, on OOD (Entity), the best-performing baseline CPT (Full) achieves 18.2% lower performance than MEND+Propagation. Even on OOD (Both), where MEND+Propagation does not observe any entity or relation in the test, MEND+Propagation is able to offer slightly better propagation than others. Interestingly, we observe that OOD (Entity) performance tends to be higher than OOD (Relation), implying that entity and relation do not share the same level of difficulty for propagation.

In Table 28, results with `Llama-3.2-3B-base-QA` shows similar pattern in Table 21, and Table 2.

Table 13: Ablation Studies of MEND+Propagation on `Controlled RippleEdit` with `Llama-3.2-1B-base-QA`. To reduce compute costs, we run MEND+Propagation (Mid-Upper), which targets Layer-`[10-12]` for editing. "Upper layer" is Layer-`[13-15(top)]`. †means the system is out-performed by MEND+Propagation (Mid-Upper) according to a paired bootstrapping test ($p = 0.05$).

| LLM-Score (↑) | In-Domain (2284) | | OOD (Entity) (1368) | | OOD (Relation) (421) | | OOD (Both) (447) | |
|---|---|---|---|---|---|---|---|---|
| | Effi. | Spec. | Effi. | Spec. | Effi. | Spec. | Effi. | Spec. |
| MEND+Propagation (Mid-Upper) | **60.8** | 91.3 | **36.0** | 85.4 | **28.4** | 87.4 | **18.3** | 84.0 |
| propagations → paraphrases | 12.4† | 91.8 | 10.5† | **93.1** | 11.8† | **93.2** | 12.9† | **89.1** |
| all tokens → answer tokens | 45.9† | 91.7 | 34.8 | 89.5 | 20.5† | 89.7 | 16.2 | 88.3 |
| Mid-Upper → Upper layers | 42.5† | **93.8** | 19.4† | 84.1 | 20.6† | 89.1 | 11.5† | 82.5 |

Table 14: Ablation studies of MEND+Propagation on `Controlled RippleEdit` with `Qwen-2.5-1.5B-base-QA`. To reduce compute costs, we run MEND+Propagation (Mid-Upper), which targets Layer-`[18-22]` for editing. "Upper layer" is Layer-`[23-27(top)]`. †means the system is out-performed by MEND+Propagation (Mid-Upper) accroding to a paired bootstrap test ($p = 0.05$).

| LLM-Score (↑) | In-Domain (2284) | | OOD (Entity) (1368) | | OOD (Relation) (421) | | OOD (Both) (447) | |
|---|---|---|---|---|---|---|---|---|
| | Effi. | Spec. | Effi. | Spec. | Effi. | Spec. | Effi. | Spec. |
| MEND+Propagation (Mid-Upper) | **56.7** | 89.5 | **30.6** | 83.0 | **28.4** | 85.7 | 14.0 | 87.9 |
| propagations → paraphrases | 10.6† | 89.9 | 9.3† | **90.4** | 12.6† | 84.6 | 10.2† | **88.3** |
| all tokens → answer tokens | 42.5† | **92.4** | 30.0 | 89.0 | 22.7† | **86.0** | **14.7** | 88.2 |
| Mid-Upper → Upper layers | 41.2† | 91.4 | 21.1† | 80.6† | 18.2† | 82.4† | 9.9† | 82.3† |

Table 15: **Stratified efficacy results on `Controlled RippleEdit`** with `Llama-3.2-1B-base-QA`. We separate the propagation questions in each test set by the criteria of whether the gold answer appear verbatim in meta-training set — denoted "In" and "Not In" respectively. We use the model's LLM-Score on multi-hop questions for measuring efficacy.

| LLM-Score (↑) | #token | OOD (Entity) | | OOD (Rel) | | OOD (Both) | |
|---|---|---|---|---|---|---|---|
| | | In (612) | Not In (756) | In (182) | Not In (239) | In (110) | Not In (337) |
| `Llama-3.2-1B-base-QA` | 0× | 13.0 | 2.4 | 16.3 | 3.3 | 35.5 | 2.9 |
| CPT (Full) | 1× | 22.2 | 12.8 | 29.6 | 4.9 | 33.7 | 6.1 |
| Meta-Aug CPT (Full) | 7× | 79.6 | 78.8 | 49.1 | 8.5 | 23.1 | 9.5 |
| Active-Reading CPT (Full) | 95× | 22.1 | 16.8 | 35.2 | 14.2 | 43.3 | 10.2 |
| MEND+Propagation | 1× | 49.1 | 24 | 63.6 | 12.4 | 62.3 | 3.9 |

# G  HYPERPARAMETERS

In Table 22, we put the hyperparameters for supervised-finetuning conducted in our study to align model output format.

In Table 24, we put the hyperparameters for meta-training MEND+Propagation and MEND. We mostly follows the default setting.

In Table 25, we put the hyperparameters for MEMIT. We mostly follows existing configurations in EasyEdit (Wang et al., 2024).

In Table 23, we put the hyperparameters for CPT baselines for both CPT (Full) and CPT (Mid-Upper).

Table 16: Efficiency Evaluation with `Llama-3.2-1B-base-QA` model on 50 examples. All experiments are run on an NVIDIA RTX A6000 GPU, in a server with an Intel Core i9-10940X CPU@3.30GHz. *: we ran 4 gradient update on the injected fact **f**, beyond which the drop in loss is marginal (see full hyperparameters in Table 23).

| | Max Memory Usage (MiB ↓) | Total Runtime (Second ↓) |
|---|---|---|
| Base Model | 6059 | 42 |
| Prepend | + 28 | + 1 |
| CPT (Full)* | + 19132 | + 920 |
| MEMIT (wikitext-103) | + 4010 | + 1291 |
| MEND (Mid-Upper) | + 7550 | + 106 |
| MEND+Propagation (Mid-Upper) | + 7542 | + 96 |
| MEND+Propagation | + 15163 | + 122 |

Table 17: Efficiency Evaluation with `Qwen-2.5-1.5B-base-QA` model on 50 examples. All experiments are run on an NVIDIA GH200 120GB, in a server with a CPU of ARM Neoverse-V2. [*]: we ran 4 gradient update on the injected fact $\mathbf{f}$, beyond which the drop in loss is marginal (see full hyperparameters in Table 23).

|  | Max Memory Usage (MiB ↓) | Total Runtime (Second ↓) |
|---|---|---|
| Base Model | 6763 | 61 |
| Prepend | + 20 | - 4 |
| CPT (Full)[*] | + 25160 | + 1442 |
| MEMIT (wikitext-103) | + 4966 | + 1059 |
| MEND (Mid-Upper) | + 8747 | + 111 |
| MEND+Propagation (Mid-Upper) | + 8741 | + 84 |
| MEND+Propagation | + 10217 | + 102 |

Table 18: Scale-up experiment of MEND+Propagation on `Controlled RippleEdit` with `Llama-3.2-1B-base-QA`. We experiment with more in-domain meta-training instances, and different sizes of hypernetwork by having dedicated hypernetworks per target weight in `Llama-3.2-1B-base-QA`. We observed that having larger training data and hypernetwork tends to improve performances on Out-of-Domain instances, but it remains challenging.

| LLM-Score (↑) | # Hypernet Param. | # train instances | In-Domain (2284) | | OOD (Entity) (1368) | | OOD (Relation) (421) | | OOD (Both) (447) | |
|---|---|---|---|---|---|---|---|---|---|---|
| | | | Effi. | Spec. | Effi. | Spec. | Effi. | Spec. | Effi. | Spec. |
| MEND+Propagation | 159M | 4K | 76.7 | 95.5 | 35.2 | 81.6 | 34.5 | 84.0 | 18.3 | 77.5 |
| | 2.8B | 30K | **97.8** | **97.1** | **42.5** | **87.2** | **41.8** | **89.5** | **20.9** | **87.8** |

# H RELATED WORK DISCUSSION

## H.1 OTHER PROPAGATION BENCHMARKS

Other benchmarks have attempted to capture knowledge propagation. DeepKnowledge (Xu et al., 2025) is a concurrent dataset testing propagation at various levels, but this dataset is not yet released at the time of development. MQuake and its improved version MQuake-Remastered (Zhong et al., 2023; 2025) aim at capturing propagation by testing whether the model is able to conduct multi-hop reasoning. In our preliminary study, we also considered a multi-hop question answering dataset for our study, but we found 100% verbatim rate from instances in MQuake-Remastered. A similar issue exists in MuSiQue (Trivedi et al., 2022) and other multi-hop question answering datasets (Yang et al., 2018). Onoe et al. (2023; 2022) study the task of learning a new entity through description (e.g., "*Dracula*"), and ask inference questions about the entity (e.g., "Dracula makes you *fear*"). CodeUpdateArena (Liu et al., 2025) tests whether the model could learn a function update in the

Table 19: Ablation Studies of MEND+Propagation on `Controlled RippleEdit` with `Llama-3.2-1B-base-QA`. To reduce compute costs, we run MEND+Propagation (Mid-Upper), which targets Layer-`[10-12]` for editing. "Upper layer" is Layer-`[13-15(top)]`. [†]means the system is out-performed by MEND+Propagation (Mid-Upper) according to a paired bootstrapping test ($p = 0.05$).

| LLM-Score (↑) | In-Domain (2284) | | OOD (Entity) (1368) | | OOD (Relation) (421) | | OOD (Both) (447) | |
|---|---|---|---|---|---|---|---|---|
| | Effi. | Spec. | Effi. | Spec. | Effi. | Spec. | Effi. | Spec. |
| MEND+Propagation (Mid-Upper) | **60.8** | 91.3 | **36.0** | 85.4 | **28.4** | 87.4 | **18.3** | 84.0 |
| propagations → paraphrases | 12.4[†] | 91.8 | 10.5[†] | **93.1** | 11.8[†] | **93.2** | 12.9[†] | **89.1** |
| all tokens → answer tokens | 45.9[†] | 91.7 | 34.8 | 89.5 | 20.5[†] | 89.7 | 16.2 | 88.3 |
| Mid-Upper → Upper layers | 42.5[†] | **93.8** | 19.4[†] | 84.1 | 20.6[†] | 89.1 | 11.5[†] | 82.5 |

Table 20: Efficiency Evaluation with `Llama-3.2-1B-base-QA` model on 50 examples. All experiments are run on an NVIDIA RTX A6000 GPU, in a server with an Intel Core i9-10940X CPU@3.30GHz. *: we ran 4 gradient update on the injected fact **f**, beyond which the drop in loss is marginal (see full hyperparameters in Table 23).

|  | Max Memory Usage (MiB ↓) | Total Runtime (Second ↓) |
|---|---|---|
| Base Model | 6059 | 42 |
| Prepend | + 28 | + 1 |
| CPT (Full)* | + 19132 | + 920 |
| MEMIT (`wikitext-103`) | + 4010 | + 1291 |
| MEND (Mid-Upper) | + 7550 | + 106 |
| MEND+Propagation (Mid-Upper) | + 7542 | + 96 |
| MEND+Propagation | + 15163 | + 122 |

Table 21: Results on `Controlled RippleEdit` with `Qwen-2.5-1.5B-base-QA`. We report the model's LLM-Score on the dataset for efficacy, and the model's performance on a collection of single-hop questions for specificity. OOD (Entity) means using ID relation with OOD entity; OOD (Relation) means using ID entity with OOD relation. Prepend is not a parametric method. †means the system is outperformed by MEND+Propagation according to a paired bootstrap test ($p = 0.05$).

| LLM-Score (↑) | In-Domain (2284) | | OOD (Entity) (1368) | | OOD (Relation) (421) | | OOD (Both) (447) | |
|---|---|---|---|---|---|---|---|---|
|  | Effi. | Spec. | Effi. | Spec. | Effi. | Spec. | Effi. | Spec. |
| `Qwen-2.5-1.5B-base-QA` | 8.0† | 91.2† | 6.8† | 89.9 | 10.5† | **87.3** | 9.1† | **91.1** |
| Prepend | 63.1 | 86.2† | **59.4** | 86.9 | **58.6** | 82.9 | **51.9** | 81.5† |
| CPT (Full) | 12.0† | 88.2† | 9.6† | 86.8 | 12.0† | 82.7 | 11.2† | 82.0† |
| Meta-Aug CPT (Full) | 85.2 | 87.5 | – | – | 32.1 | 73.6 | 18.2 | 73.8 |
| Active Reading CPT (Full) | 16.1 | 80 | 15.2 | 79.9 | 19.5 | 73.8 | 14.3 | 75.3 |
| MEMIT (`wikitext-103`) | 16.0† | 91.3† | 16.1† | 90.1 | 13.9† | 87.2 | 9.6† | 90.3 |
| MEMIT (`Ctrl RippleEdit`) | 11.6† | 91.2† | 12.6† | 90.0 | 10.3† | 86.6 | 10.1† | 89.7 |
| MEND (with standard config) | 12.3† | 87.1† | 9.9† | 88.2 | 11.1† | 83.5 | 10.9† | 86.2 |
| MEND (Mid-Upper) | 9.1† | 58.3† | 8.9† | 56.6† | 4.8† | 61.4† | 5.2† | 69.4† |
| MEND+Propagation (Mid-Upper) | 56.7† | 89.5† | 30.6† | 83.0 | 28.4† | 85.7 | 14.0† | 87.9 |
| MEND+Propagation | **64.0** | **93.6** | 34.7 | 83.0 | 33.3 | 84.8 | 17.7 | 85.8 |

docstring difference and apply the updated function in program synthesis. ECLeKTic (Goldman et al., 2025) focuses on cross-lingual knowledge transfer.

### H.2 CONTINUED PRETRAINING

Continued pretraining (CPT) on documents to be injected serves as a strong baseline in these scenarios. A line of work (Padmanabhan et al., 2023; Akyürek et al., 2024) proposes to improve knowledge propagation in CPT by modifying data scenarios or learning objectives. Yao et al. (2025) uses circuit analysis to arrive at the template for data augmentation. Jiang et al. (2024) finds instruction-tuning LMs on question-answering pairs prior to CPT is beneficial for knowledge injection. Yang et al. (2024) proposes to synthesize large-scale data from the document to be injected and perform CPT on those documents, showing improved propagation. Unlike this line of work, hypernetwork-based methods do not synthesize additional data at test time.

## I COMPUTATIONAL RESOURCES

We conducted experiments with `Llama-3.2-1B-base` primarily on a server with NVIDIA A40 48GB GPUs and an AMD EPYC 7413 24-Core Processor. For larger models, our experiments were conducted on a server with NVIDIA GH200 120GB and ARM Neoverse-V2.

Table 22: Hyperparameters used for Supervised Fine-Tuning (SFT). The same set of parameters was used for `Llama-3.2-1B-base`, `Qwen-2.5-1.5B-base`, and `Llama-3.2-3B-base` (suffixed by `-QA`).

<table>
<tr><td colspan="2">(a) SFT on TriviaQA.rc.</td><td></td><td colspan="2">(b) SFT on `Controlled RippleEdit`.</td></tr>
<tr><td>Hyperparamter</td><td>Value</td><td></td><td>Hyperparamter</td><td>Value</td></tr>
<tr><td>Learning rate</td><td>1e-5</td><td></td><td>Learning rate</td><td>2e-6</td></tr>
<tr><td>Scheduler</td><td>linear</td><td></td><td>Scheduler</td><td>linear</td></tr>
<tr><td>Epoch</td><td>2</td><td></td><td>Epoch</td><td>2</td></tr>
<tr><td>Max seq. length</td><td>256</td><td></td><td>Max seq. length</td><td>256</td></tr>
<tr><td>Batch size</td><td>128</td><td></td><td>Batch size</td><td>10</td></tr>
<tr><td>Weight decay</td><td>0.1</td><td></td><td>Weight decay</td><td>0.1</td></tr>
<tr><td>Max Gradient Norm</td><td>1.0</td><td></td><td>Max Gradient Norm</td><td>1.0</td></tr>
<tr><td>WarmUp ratio</td><td>0.03</td><td></td><td>WarmUp ratio</td><td>0.03</td></tr>
<tr><td>Optimizer</td><td>AdamW</td><td></td><td>Optimizer</td><td>AdamW</td></tr>
</table>

Table 23: Hyperparameters used for Continue Pretraining baselines, CPT (Full) and CPT (Mid-Upper), when injecting one fact $\mathbf{f}$.

| Hyperparamter | Value |
| --- | --- |
| Learning rate | 1e-5 |
| Scheduler | linear |
| Epoch | 4 |
| Max seq. length | 1024 |
| Batch size | 1 |
| Weight decay | 0.1 |
| Max Gradient Norm | 1.0 |
| Optimizer | AdamW |

Table 24: Hyperparameters used for MEND+Propagation and MEND.

(a) Hyperparameters for training MEND+Propagation and MEND.

| Hyperparameter | Value |
| --- | --- |
| $c_{edit}$ | 0.1 |
| learning rate to learn test-time learning rate $\alpha_\ell$ | 0.0001 |
| Learning rate for hypernetwork weight $\phi$ | 1.0e-06 |
| Batch size (after gradient accumulation) | 10 |
| Validation step | 100 |
| Early stop patience (# steps) | 2000 |
| Maximum training step | 1000000 |
| Optimizer | Adam |

(b) Hyperparameters for hypernetwork (MLP) in MEND+Propagation and MEND.

| Hyperparameter | Value |
| --- | --- |
| Activation | ReLU |
| # hidden | 1 |
| # hidden dim | 1920 |
| # parameter sharing | False |

(c) Target MLP layers used for various comparison system

| Base Model | Total # layers | Comparison system | Layer indices (min: 0) |
| --- | --- | --- | --- |
| `Llama-3.2-1B-base` | 16 | MEND+Propagation | 4-15 |
| | | MEND+Propagation (Mid-Upper)/ MEND (Mid-Upper) | 10-12 |
| `Qwen2.5-1.5B-base` | 28 | MEND+Propagation | 13-27 |
| `Llama-3.2-3B-base` | 28 | MEND+Propagation | 15-27 |

Table 25: Hyperparameters used for MEMIT.

(a) For `Llama-3.2-1B-base`

| Hyperparameter | Value |
|---|---|
| Target layer | [1, 2, 3, 4, 5] |
| rewrite_module_tmp | "layers.{}.mlp.down_proj" |
| clamp_norm_factor | 0.75 |
| fact_token | "subject_last" |
| v_num_grad_steps | 20 |
| v_lr | 5e-1 |
| v_loss_layer | 15 |
| v_weight_decay | 0.5 |
| kl_factor | 0.0625 |
| mom2_adjustment | true |
| mom2_update_weight | 20000 |
| mom2_n_samples | 100000 |

(b) For `Qwen-2.5-1.5B-base`

| Hyperparameter | Value |
|---|---|
| Target layer | [4, 5, 6, 7, 8] |
| rewrite_module_tmp | "layers.{}.mlp.down_proj" |
| clamp_norm_factor | 4 |
| fact_token | "subject_last" |
| v_num_grad_steps | 25 |
| v_lr | 5e-1 |
| v_loss_layer | 27 |
| v_weight_decay | 1e-3 |
| kl_factor | 0.0625 |
| mom2_adjustment | true |
| mom2_update_weight | 15000 |
| mom2_n_samples | 100000 |

Table 26: **Exact Match (EM) Results on `RippleEdit` with `Llama-3.2-1B-base-QA`**. We report the total number of test queries in brackets. Prepend is not a parametric method. The other metric (LLM-Score) is reported in Table 1 in the main paper.

| EM (↑) | Efficacy | | Specificity | |
|---|---|---|---|---|
| | Verbatim | Non-Verbatim | Verbatim | Non-Verbatim |
| | (1373) | (1586) | (165) | (2099) |
| `Llama-3.2-1B-base-QA` | 17.0 | 4.0 | 90.9 | 23.2 |
| Prepend | 36.0 | 12.4 | 94.5 | 21.6 |
| CPT (Full) | 87.8 | 3.4 | **99.4** | 17.3 |
| CPT (Mid-Upper) | 48.7 | 4.0 | 93.3 | 24.1 |
| MEMIT (`wikitext-103`) | 21.1 | 5.6 | 93.3 | 24.1 |
| MEMIT (`RippleEdit`) | 26.6 | 5.9 | 98.2 | 19.3 |
| MEND (with standard config) | 72.7 | 3.0 | 98.2 | 21.3 |
| MEND (Mid-Upper) | 69.7 | 3.1 | 97.0 | 17.8 |
| MEND+Propagation (Mid-Upper) | 73.8 | 14.9 | 97.6 | 31.8 |
| MEND+Propagation | **78.7** | **17.3** | 95.2 | **35.1** |

Table 27: **Results on `RippleEdit` with `Llama-3.2-1B-base-QA`**. Performances are reported in the format of Exact Match (EM) / LLM-Score. We notice the EM and LLM-Score strongly disagree with each other on Forgetfulness (FN); after spotchecking, we found EM is high because one of the valid answers $a \in \mathcal{A}_i$ is a substring of the propagation question $\mathbf{q}_i$. Prepend is not a parametric method.

| EM / LLM-Score (↑) | Efficacy | | | | Specificity | |
|---|---|---|---|---|---|---|
| | LG | CI | CII | SA | RS | FN |
| | (230) | (1679) | (273) | (777) | (1982) | (282) |
| `Llama-3.2-1B-base-QA` | 13.0/13.5 | 13.0/11.0 | 4.4/9.3 | 4.6/8.2 | 24.9/29.0 | 51.1/10.4 |
| Prepend | 20.0/31.9 | 21.1/24.9 | 18.3/22.6 | 30.9/39.2 | 23.3/30.0 | 52.5/13.6 |
| CPT (Full) | 16.1/11.4 | 12.7/10.4 | 93.8/89.3 | 97.0/93.0 | 19.9/17.8 | 47.5/3.3 |
| CPT (Mid-Upper) | 13.9/15.8 | 13.3/12.0 | 32.6/32.2 | 50.1/51.7 | 26.4/28.0 | 48.6/10.9 |
| Active-Reading CPT (Full) | 36.5/29.9 | 13.2/12.7 | 93.8/95.1 | 97.7/93.8 | 20.7/19.7 | 48.9/5.4 |
| MEMIT (`wikitext-103`) | 14.3/13.8 | 14.5/14.6 | 7.3/11.6 | 10.6/16.2 | 24.1/26.3 | 49.6/7.9 |
| MEMIT (`RippleEdit`) | 14.3/13.3 | 14.8/14.8 | 7.7/13.9 | 20.2/24.9 | 21.6/23.5 | 48.9/7.3 |
| MEND (with standard config) | 14.8/11.7 | 12.1/10.2 | 68.9/69.8 | 79.9/80.8 | 24.0/25.8 | 47.5/8.4 |
| MEND (Mid-Upper) | 13.5/13.8 | 12.4/10.8 | 59.0/64.1 | 77.9/79.2 | 20.1/23.6 | 47.5/8.1 |
| MEND+Propagation (Mid-Upper) | 27.0/12.8 | 22.9/25.9 | 72.5/74.3 | 77.7/79.3 | 33.3/33.1 | 59.9/21.5 |
| MEND+Propagation | 30.9/25.0 | 25.3/27.7 | 83.5/85.7 | 81.3/82.1 | 35.7/35.6 | 65.6/27.3 |

Table 28: Results on `Controlled RippleEdit` with `Llama-3.2-3B-base-QA`. We use the model's LLM-Score on multi-hop questions for efficacy, and the model's performance on single-hop questions for specificity. OOD (Entity) means using ID relation with OOD entity; OOD (Relation) means using ID entity with OOD relation. Prepend is not a parametric method.

| LLM-Score (↑) | In-Domain (2284) | | OOD(Entity) (1368) | | OOD(Relation) (421) | | OOD(Both) (447) | |
|---|---|---|---|---|---|---|---|---|
| | Effi. | Spec. | Effi. | Spec. | Effi. | Spec. | Effi. | Spec. |
| `Llama-3.2-3B-base-QA` | 8.1 | 91.8 | 6.9 | 93.0 | 8.1 | 92.4 | 6.5 | 93.8 |
| Prepend | 66.1 | 90.3 | 62.5 | 92.1 | 61.3 | 90.3 | 52.5 | 91.6 |
| CPT (Full) | 18.4 | 86.2 | 16.8 | 86.0 | 16.1 | 86.7 | 12.7 | 82.7 |
| MEND+Propagation | 69.9 | 94.6 | 42.4 | 89.8 | 34.0 | 93.2 | 19.2 | 89.6 |

Though the runtime varies depending on the datasets, the meta-training of hyper networks typically takes around 10 hours, or as little as 4 hours for some experiments.

