# OpenReview forum: "Propagating Knowledge in LLMs with Hypernetworks"
_ICLR.cc/2026/Conference — ICLR 2026 Conference Withdrawn Submission_

### Official Review · Reviewer_JxeR · 2025-10-26

**Soundness:** 1
**Presentation:** 3
**Contribution:** 1
**Rating:** 2
**Confidence:** 4

**Summary:**

This paper investigates the challenge of an edited model to reason about and apply newly injected facts beyond verbatim recall. While prior knowledge editing techniques (MEND, MEMIT, ROME) enable efficient and localized parameter updates via hypernetworks, they largely fail to propagate knowledge to related, multi-hop inferences. The authors propose a simple but effective modification to hypernetwork-based editing: they train the hypernetwork not only on direct recall of injected facts but also on propagation questions that require reasoning over those facts. This change directly optimizes the hypernetwork to modify gradients for propagation, rather than memorization.

Empirically, the paper shows that this objective improves propagation performance on the RippleEdit benchmark. To further analyze generalization, the authors introduce Controlled RippleEdit, a synthetic dataset that isolates confounding factors such as entity familiarity and relation overlap. Their results demonstrate improvements in the in-domain setting, though the OOD setting remains challenging (performance drops from 76.7% to 18.3%). Finally, they extend their training objective to multi-edit scenarios using RLEdit, showing that their propagation-aware hypernetwork scales comparably or better than continual-pretraining baselines.

**Strengths:**

The paper addresses an important and timely problem in the field of knowledge editing: the limited ability of edited LLMs to propagate newly injected knowledge to related reasoning or multi-hop questions. This issue represents one of the most pressing limitations of current editing methods and has direct implications for maintaining up-to-date, reliable LLMs without full retraining. Tackling this challenge is therefore of high practical and scientific significance, and the paper makes clear progress toward understanding and mitigating it.

In terms of clarity, the paper is very well structured and accessible. The motivation is clearly articulated, the methodology is logically presented, and the visualizations are thoughtfully designed to convey the mechanism of propagation-aware editing. These choices make the technical contributions easy to follow.

**Weaknesses:**

While the paper tackles an important problem, it has several notable weaknesses that limit the strength and generality of its conclusions.

The most significant limitation of the paper is a conceptual flaw in its experimental setup that fundamentally undermines the main claim of improved knowledge propagation. The proposed approach explicitly incorporates the multi-hop propagation questions (which are supposed to serve as the evaluation set) into the hypernetwork’s training objective. This design leads to substantial data leakage, as the model is effectively trained on the same questions it is later tested on. Consequently, the reported improvements cannot be interpreted as evidence of genuine reasoning or propagation ability, but rather as a by-product of exposure to the test items during training. This issue becomes even clearer in the results reported in Table 2, where the authors evaluate their method in what they term an “out-of-domain” setting. However, this setup does not constitute a true OOD scenario, but instead, it corresponds to a standard unseen-test configuration, in which the evaluation examples are simply withheld from training. Once this overlap between training and evaluation is removed, the performance of the proposed method drops sharply, revealing that the earlier gains stem largely from data leakage rather than from an improved ability to propagate knowledge.

The paper’s experimental scope is narrow, relying on a single dataset (RippleEdit and its controlled variant) and a single, relatively small base model that is not instruction-tuned. Evaluating only one model configuration prevents a robust assessment of whether the proposed propagation-aware hypernetwork scales to larger or more realistic setups (e.g., 4–8 B parameter instruct models commonly used in editing studies).

The choice of baselines could be improved: the continued pretraining comparison is outdated, given that recent work shows parameter-efficient fine-tuning combined with model merging offers far stronger and more stable editing performance [1].

The interpretation of the results in Table 1 is inconsistent. The caption claims improvements in both verbatim and non-verbatim cases, yet the method underperforms Active-Reading CPT and CPT (Full) in the verbatim setting and is only on par with prepend in the non-verbatim case. More generally, the empirical results are not uniformly convincing: across Tables 1 and 2, the proposed method outperforms baselines only in select configurations, making it difficult to support the broad claim of a generalizable improvement in knowledge propagation (especially given that all results stem from only a single model).

While Section 2.3 mentions scalability in terms of multiple simultaneous edits, it omits discussion of the lifelong or sequential editing scenario, which is more practically relevant and actively explored in recent work such as WISE [2] and WikiBigEdit [3]. The authors should at least mention this line of work here.

[1] Roth et al. (2024), A Practitioner’s Guide to Continual Multimodal Pretraining

[2] Wang et al. (2024), WISE: Rethinking the Knowledge Memory for Lifelong Model Editing of Large Language Models

[3] Thede et al. (2025), WikiBigEdit: Understanding the Limits of Lifelong Knowledge Editing in LLMs

**Questions:**

1) How do the authors ensure that the reported improvements are not simply due to data leakage or memorization of test items?
2) The paper refers to the split in Table 2 as “out-of-domain,” but it appears to correspond to a standard unseen-test configuration rather than a true OOD scenario. Could the authors elaborate on how this split was constructed and whether any relations or entities overlap between training and evaluation?
3) Do the authors plan to validate their approach on larger or instruction-tuned models (e.g., 4–8 B parameter range) and on additional benchmarks that capture different types of propagation? This would help assess the method’s generality beyond a single dataset and model.
4) Why was naive continued pretraining chosen as the main baseline instead of more competitive and parameter-efficient alternatives, such as fine-tuning plus model merging?

---

> ### Author Response · Authors · 2025-11-23
> **Response to reviewer (1/3)**
>
> > This design leads to substantial data leakage, as the model is effectively trained on the same questions it is later tested on. Consequently, the reported improvements cannot be interpreted as evidence of genuine reasoning or propagation ability, but rather as a by-product of exposure to the test items during training.
>
> We don’t understand the criticism here. There is no setting in which the actual test instances are leaked into the training data. Our results are stratified based on whether the following appear in the training data: (1) entity + relation; (2) just entity; (3) just relation; (4) neither. We think this makes it quite explicit what is in the data.
>
> There is some potential overlap in the answers (e.g., the meta-train data might contain the answer “US” for a question about someone’s country of origin, which could be in the test data for another location query). But again, we don’t think this can be characterized as data leakage.
>
>
>
> > This issue becomes even clearer in the results reported in Table 2, where the authors evaluate their method in what they term an “out-of-domain” setting.
>
> Our results in Table 2 exactly shows our meta-learned operator is able to generalize to OOD, albeit with a performance drop.
>
> > However, this setup does not constitute a true OOD scenario, but instead, it corresponds to a standard unseen-test configuration
>
> No, we strongly disagree. Since the examples in OOD scenarios are generated from OOD entities and/or relations that never appear in a meta-training set (i.e. in-domain), the distribution is not I.I.D. with the training set. Our In-Domain test set is IID. See more details below.
>
> > The paper refers to the split in Table 2 as “out-of-domain,” but it appears to correspond to a standard unseen-test configuration rather than a true OOD scenario. Could the authors elaborate on how this split was constructed and whether any relations or entities overlap between training and evaluation?
>
> We discuss OOD test set construction in Line 344-349.
>
> Our Controlled RippleEdit dataset is constructed by using real-world entities, entity-specific relations, and entity specific templates:
> 1. We split entities into in-domain entities and OOD entities; and relations into in-domain relations and OOD relations;
> 2. We generate the meta-training set by only using in-domain entities and in-domain relations. And randomly generated names for fake entities.
>
> |                    | In-Domain entities | OOD entities |
> |--------------------|--------------------|--------------|
> | In-Domain relation | In-Domain test     | OOD (Entity) |
> | OOD relations      | OOD (Relation)     | OOD (Both)   |
>
>
> > The paper’s experimental scope is narrow, relying on a single dataset (RippleEdit and its controlled variant)
>
> We note that Controlled RippleEdit is distinct from RippleEdit in a few ways:
> 1. It features a different editing setup, where longer blocks of text are injected
> 2. It’s stratified by relations and entities
>
> In addition, RippleEdit itself contains several propagation types and multiple subsets based on entity popularity. Despite the similar naming, we view these two datasets as testing quite a broad range of propagation phenomena. RippleEdit has been used in several prior papers as the chief basis for evaluation [1, 2].
>
> [1] RIPPLECOT: Amplifying Ripple Effect of Knowledge Editing in Language
> Models via Chain-of-Thought In-Context Learning EMNLP-Finding 2024 Zihao Zhao, Yuchen Yang, Yijiang Li, Yinzhi Cao
>
> [2] SAKE: Steering Activations for Knowledge Editing ACL 2025 Marco Scialanga, Thibault Laugel, Vincent Grari, Marcin Detyniecki
>
> > Evaluating only one model configuration prevents a robust assessment of whether the proposed propagation-aware hypernetwork scales to larger or more realistic setups (e.g., 4–8 B parameter instruct models commonly used in editing studies).
>
> We disagree that we only test on a single model configuration. In main paper, we leave pointers to more results at Line 413:
> 1. See Table 21 in Appendix for results of Qwen-2.5-1B-base-QA on Controlled RippleEdit
> 2. See Table 28 in Appendix for results of Llama-3.2-3B-base-QA on Controlled RippleEdit
> 3. See our response below for results of Llama-3.1-8B-base-QA on Controlled RippleEdit.

---

> > ### Comment · Reviewer_JxeR · 2025-11-24
> > **Data Leakage and OOD Setting**
> >
> > Thank you for the detailed rebuttal and for the clarifications regarding the construction of the meta-training and evaluation splits. After revisiting the paper, I would like to correct one aspect of my original review: there is no direct data leakage of test questions into the training set. My initial wording suggested this, and that was based on a misunderstanding of the setup.
> >
> > That said, two central concerns from my earlier review remain unresolved:
> >
> > **Over-specialization to the propagation question format.** While the test instances themselves are not leaked, the hypernetwork is trained directly on propagation questions of the exact same structural form as those used for evaluation. This means the model is learning the specific multi-hop reasoning templates and query structure it is later evaluated on. As a result, it remains unclear whether the method would generalize to different question styles, unseen templates, or different evaluation tasks. The current experiments do not provide evidence of broader or more abstract propagation ability beyond the specific patterns seen during meta-training.
> >
> > **This evaluation is not “OOD” in the classical sense; rather, it tests generalization to unseen entities/relations.** In standard literature on distribution shift [1,2,3], an OOD setting typically requires a change in the underlying data-generating distribution, such as new input structures, new reasoning patterns, new template families, or covariate shifts. In this paper, however, the “OOD” splits preserve the identical question templates, multi-hop structure, and reasoning format as the meta-training distribution; only the specific entity or relation identifiers differ. This corresponds more closely to unseen-entity or unseen-relation generalization, not to a structural or distributional OOD scenario in the classical sense. Apart from the naming, the more important issue is the practical implication of the performance drop. Even under this relatively mild form of generalization (unseen entities or relations), performance decreases sharply (e.g., from 76.7% to 18.3% in OOD-Both). Combined with the scaling ablations, which show that additional meta-training data or larger hypernetworks do not close this gap, the results suggest that the method may require substantial coverage of entities and relations during meta-training in order to perform reliably. For real-world factual editing, where entities and relations number in the millions and new ones appear continually, this raises significant concerns about scalability and practicality. Without stronger generalization to unseen entities and relations, the hypernetwork may require frequent retraining or extremely broad meta-training coverage to remain effective.
> >
> > [1] Hendrycks and Dietterich (2019) Benchmarking Neural Network Robustness to Common Corruptions and Perturbations
> > [2] Arjovsky et al. (2019) Invariant Risk Minimization
> > [3] Koh et al. (2021) WILDS: A Benchmark of In-the-Wild Distribution Shifts.

---

> > > ### Comment · Reviewer_JxeR · 2025-11-24
> > > **Experimental Scope**
> > >
> > > Thank you for the clarifications. I see the authors’ perspective that RippleEdit and Controlled RippleEdit together cover a variety of propagation settings, different fact formats, entity and relation stratifications, and multiple propagation types. Nonetheless, I still believe the experimental scope remains somewhat limited. The paper would benefit from demonstrating propagation-oriented hypernetwork training in additional settings or datasets, although I recognize that such experiments are computationally demanding.
> > > Regarding the evaluation on different base models, thank you for pointing me to the appendix results. I had originally missed these. To improve clarity and visibility for readers, I would encourage the authors to either include these experiments directly in the main paper or clearly reference them in the relevant discussion. The current phrasing (“In Appendix F, we report further results…”) does not make it obvious that additional model configurations are evaluated.

---

> > > > ### Comment · Reviewer_JxeR · 2025-11-24
> > > > **Experiments on larger models and instruct models.**
> > > >
> > > > Thank you for providing the additional experiments on Llama-3.2-1B-Instruct and Llama-3.1-8B-base. Including these results will strengthen the submission and better demonstrate the method’s applicability beyond the 1B base model.
> > > >
> > > > Regarding Llama-3.2-1B-Instruct, the drop in in-domain efficacy compared to the base model is striking, especially since the “OOD” performance remains relatively similar. Do the authors have any intuition for why instruction tuning disproportionately harms in-domain propagation but not unseen-entity or unseen-relation generalization?
> > > >
> > > > For the 8B results, the prepend baseline performs unexpectedly well, particularly in the OOD settings. How do the authors interpret this? If retrieval-augmented prompting already works so effectively (even for unseen entities and relations) what is the main argument for editing rather than simply providing the updated fact in context?
> > > >
> > > > Finally, I appreciate the authors’ transparency about the difficulty of scaling to larger models. What is their assessment of extending this method to even larger models (e.g., 30B–80B)? Should we expect the approach to remain viable with appropriate modifications, or are there fundamental obstacles to propagation-aware hypernetwork editing at that scale?

---

> > > > > ### Comment · Reviewer_JxeR · 2025-11-24
> > > > > **CPT Baseline**
> > > > >
> > > > > Thank you for the clarifications regarding the baselines. To avoid misunderstanding, I did not mean to refer to a specific “model-merging knowledge editing” method. My point was that model merging can serve as a strong and scalable alternative to full continued pretraining, especially for larger models where full CPT becomes prohibitively expensive. A simple LoRA fine-tuning step followed by merging back into the base model would provide a more competitive CPT baseline than naive full-parameter training, while remaining computationally feasible even at 8B scale and beyond.

---

> > > > > > ### Comment · Reviewer_JxeR · 2025-11-24
> > > > > > **General**
> > > > > >
> > > > > > I would like to thank the authors for their extensive and thoughtful rebuttal. Several of my earlier questions have been clarified, and the rebuttal helped resolve some misunderstandings on my end. I have therefore revised my score accordingly.
> > > > > >
> > > > > > The main concerns that remain for me are the following:
> > > > > > * Generalization to unseen entities and relations: The method appears to struggle when either the entity or the relation is not included in the meta-training, raising questions about its robustness in real-world settings with large, continually expanding knowledge bases.
> > > > > > * Specialization to a specific task setting: The approach seems tightly coupled to the particular propagation-style queries used during meta-training, and it is still unclear how well the method would generalize to different question styles, reasoning formats, or evaluation tasks beyond this template.
> > > > > > * (Minor) Scalability to large models: While I appreciate the transparency regarding the challenges of scaling, it remains uncertain whether the method can be applied reliably to substantially larger models.

---

> > > > > > > ### Author Response · Authors · 2025-11-25
> > > > > > > **Response to reviewer's followup**
> > > > > > >
> > > > > > > Thanks for following up!
> > > > > > >
> > > > > > > > Generalization to unseen entities and relations
> > > > > > >
> > > > > > > We agree that a worthy goal for follow-on work is to achieve better propagation OOD. However, we still think the ability to propagate “in-domain” is valuable: we see this as a demonstration of this technique that could be scaled up to cover domains of interest.
> > > > > > >
> > > > > > > > A simple LoRA fine-tuning step followed by merging back into the base model would provide a more competitive CPT baseline than naive full-parameter training, while remaining computationally feasible even at 8B scale and beyond.
> > > > > > >
> > > > > > > Our new results with the 8B model include an experiment using LoRA for CPT, which does not perform particularly well. As for merging with the base model, that’s possible, but our intuition is that that would primarily mitigate forgetting and improve generalization, not actually improve the propagation of the learned knowledge.
> > > > > > >
> > > > > > > > Do the authors have any intuition for why instruction tuning disproportionately harms in-domain propagation but not unseen-entity or unseen-relation generalization?
> > > > > > >
> > > > > > > We do not have good intuition here, but we do note that instruction-tuned models are hard to edit, where CPT baselines achieve near-zero performance. This fact means that the performance regime is just somewhat different overall.
> > > > > > >
> > > > > > > > For the 8B results, the prepend baseline performs unexpectedly well, particularly in the OOD settings. How do the authors interpret this? If retrieval-augmented prompting already works so effectively (even for unseen entities and relations) what is the main argument for editing rather than simply providing the updated fact in context?
> > > > > > >
> > > > > > > A bigger model is expected to have better instruction following capability or stronger in learning in-context. We emphasize that prepend is not really a baseline, more of an orthogonal method meant to provide context. The value of our work is to explore a different line of approach from RAG that doesn’t involve adding context at test time.
> > > > > > >
> > > > > > > > Should we expect the approach to remain viable with appropriate modifications, or are there fundamental obstacles to propagation-aware hypernetwork editing at that scale?
> > > > > > >
> > > > > > > Yes, our approach should remain viable.
> > > > > > >
> > > > > > > > Nonetheless, I still believe the experimental scope remains somewhat limited. The paper would benefit from demonstrating propagation-oriented hypernetwork training in additional settings or datasets, although I recognize that such experiments are computationally demanding.
> > > > > > >
> > > > > > > We agree this would be valuable to show, but are bound by the availability of existing datasets. Past work on knowledge editing has largely focused on entity-centric data, and the extension to Controlled RippleEdit in this paper already required additional modification.
> > > > > > >
> > > > > > > > To improve clarity and visibility for readers, I would encourage the authors to either include these experiments directly in the main paper or clearly reference them in the relevant discussion.
> > > > > > >
> > > > > > > Thanks for the suggestion. We will use the additional page to do so.

---

> > > > > > > > ### Comment · Reviewer_JxeR · 2025-11-26
> > > > > > > >
> > > > > > > > Thank you for the additional clarifications. I see the authors’ argument that strong in-domain propagation is still valuable. However, if the method is primarily intended for settings where the entities and relations are known in advance, I think the paper would benefit from framing the scope more narrowly and explicitly. As it stands, the current presentation suggests broader applicability, while the experimental results indicate that robust propagation beyond the meta-training domain remains a substantive challenge.
> > > > > > > >
> > > > > > > > Regarding the prepend baseline and the role of RAG-style methods, I agree that editing and retrieval offer different trade-offs. That said, in many practical scenarios, the cost of retrieval and adding context is low, making RAG a competitive alternative. I believe the paper would benefit from a brief mention of the trade-offs between in-weight editing and retrieval over external knowledge bases.
> > > > > > > >
> > > > > > > > Additionally, the initial results on the 8B scale emphasize the importance of including "larger" models in future evaluations. While I understand the computational constraints, models in the 1–3B range often behave quite differently from 8B+ models, and the prepend results illustrate that conclusions drawn from small models may not generalize upward. Explicitly discussing these limitations and integrating the larger-model results into the main text would help set accurate expectations for the method’s practical applicability.

---

> ### Author Response · Authors · 2025-11-23
> **Response to reviewer (2/3)**
>
> > Do the authors plan to validate their approach on larger or instruction-tuned models (e.g., 4–8 B parameter range)
>
> The test our method on both the Llama-3.2-1B-Instruct and Llama-3.1-8B-base and show our method shows similar trends over test sets as existing results.
>
> **Llama-3.2-1B-Instruct**
>
> |                       |                  | In-Domain |             | Out-of-Domain (Entity) |             | Out-of-Domain (Relation) |             | Out-of-Domain (Both) |             |
> |-----------------------|------------------|-----------|-------------|------------------------|-------------|--------------------------|-------------|----------------------|-------------|
> |                       | Method           | Efficacy  | Specificity | Efficacy               | Specificity | Efficacy                 | Specificity | Efficacy             | Specificity |
> | Llama-3.2-1B-QA       | Base             | 8.3       | 94.7        | 7.1                    | 94.3        | 8.9                      | 94.2        | 10.9                 | 90.7        |
> |                       | CPT              | 18.1      | 80.2        | 17.0                   | 79.9        | 15.6                     | 79.3        | 12.9                 | 71.1        |
> |                       | MEND+Propagation | **76.7**  | 95.5        | **35.2**               | 81.6        | **34.5**                 | 84.0        | **18.3**             | 77.5        |
> | Llama-3.2-1B-Instruct | Base             | 0.5       | 80.6        | 0.3                    | 78.4        | 0.3                      | 83.9        | 0.5                  | 88.7        |
> |                       | CPT              | 1         | 76.4        | 0.7                    | 75.3        | 0.8                      | 78.8        | 0.5                  | 79.5        |
> |                       | MEND+Propagation | **45.7**  | 87.1        | **35.5**               | 84.1        | **26.3**                 | 83.1        | **18.1**             | 84.9        |
>
>
>
> **Llama-3.1-8B-base**
>
> We do notice it’s hard to scale to larger sizes like 8B. Due to time and budget constraints, we use LoRA (`all-linear`) for CPT baselines and target 2 MLP layers (we edited 12 layers for 1B model) for our “MEND+Propagation” method. For LoRA, we adjust the learning rate so that it reaches a similar loss as Full Finetuning. In the table below, we show our method is still effective to enable the model to answer propagation questions (single edit) in OOD (Entity) and OOD (Relation)
>
> |                                  |        In-Domain        |             |       OOD (Entity)      |             |     OOD (Relation)     |             |       OOD (Both)       |             |
> |----------------------------------|:-----------------------:|:-----------:|:-----------------------:|:-----------:|:----------------------:|:-----------:|:----------------------:|:-----------:|
> | Llama-3.2-8B-base-QA Single Edit | Overall (2284) Accuracy | Specificity | Overall (1368) Accuracy | Specificity | Overall (421) Accuracy | Specificity | Overall (447) Accuracy | Specificity |
> | Base                             |           6.8           |     91.6    |           6.3           |     89.5    |          10.3          |     93.4    |           9.9          |     92.5    |
> | Prepend                          |           74.7          |     92.4    |           73.2          |      90     |          73.8          |     91.1    |          67.5          |      94     |
> | CPT (LoRA)                       |           15.4          |     90.4    |           13.8          |     88.3    |          16.7          |      92     |          16.6          |     89.2    |
> | Active-Reading CPT (LoRA)        |           19.6          |     80.2    |           17.7          |     81.3    |          22.4          |     79.4    |        **19.6**        |     76.7    |
> | MEND+Propagation                 |         **65.2**        |     92.9    |          **41**         |     87.8    |        **30.4**        |     90.7    |          15.4          |     87.9    |
>
>
> > Why do we focus on the base model?
>
> It is easier to inject knowledge to base models when the learning rate is still high. Prior work also uses a base model to study the learning of factual knowledge [1,2,3].  Anecdotally, it is difficult to inject knowledge into a model (via CPT or other means) after a long instruction training phase.
>
>
> [1] How Do Large Language Models Acquire Factual Knowledge During Pretraining?; NeurIPS 2024, Hoyeon Chang, Jinho Park, Seonghyeon Ye, Sohee Yang, Youngkyung Seo, Du-Seong Chang, Minjoon Seo
>
> [2] Memorization without overfitting: Analyzing the training dynamics of large language models; NeurIPS 2022 Kushal Tirumala, Aram H. Markosyan, Luke Zettlemoyer, Armen Aghajanyan
>
> [3] Dissecting recall of factual associations in auto-regressive language models; EMNLP 2023, Mor Geva, Jasmijn Bastings, Katja Filippova, Amir Globerson

---

> ### Author Response · Authors · 2025-11-23
> **Response to reviewer (3/3)**
>
> > The choice of baselines could be improved: the continued pretraining comparison is outdated
>
> We disagree. The Adaptive Reading [1] baseline is the latest method (2025) that we know and they claim they out-perform other recent baseline like EntiGraph[2]
>
> [1] Learning Facts at Scale with Active Reading; arxiv Jessy Lin, Vincent-Pierre Berges, Xilun Chen, Wen-Tau Yih, Gargi Ghosh, Barlas Oğuz
>
> [2] Synthetic continued pretraining; ICLR 2025 Zitong Yang, Neil Band, Shuangping Li, Emmanuel Candès, Tatsunori Hashimoto
>
> > While Section 2.3 mentions scalability in terms of multiple simultaneous edits, it omits discussion of the lifelong or sequential editing scenario
> Our multi-edit setting is the lifelong editing setting, where we inject 5 facts in each turn of edits (See Line 431-452). In revision, we will add some discussion regarding recent work on multi-edit methods like the reviewer suggested.
>
> > The interpretation of the results in Table 1 is inconsistent.
>
> Thanks for the catch. Our caption regarding verbatim is inaccurate and was compared to an old set of baselines. We will fix it.
>
> > More generally, the empirical results are not uniformly convincing: across Tables 1 and 2, the proposed method outperforms baselines only in select configurations…
>
> First, we note that specificity measurement is not our target metric for success; specificity is meant to measure whether the model significantly degrades on other capabilities.
>
> Secondly, we want to highlight in Table 1, the verbatim efficacy can be easily gamed whereas non-verbatim efficacy is not. For instance, since we are dealing with a single-edit scenario, a non-brain method that works perfectly on verbatim test questions is just to regurgitate the “output” component during injection (we note MEMIT and MEND consume fact in input-output format).
>
> As a result, our select configuration – non-verbatim efficacy (one column in Table 1 and all columns in Table 2) is the right metric to focus interpretation on. And here we do show improvements over the relevant baselines.
>
> > How do the authors ensure that the reported improvements are not simply due to data leakage or memorization of test items?
>
> We don’t understand the criticism about data leakage here. There is no setting in which the actual test instances are leaked into the training data. Our results are stratified based on whether the following appear in the training data: (1) entity + relation; (2) just entity; (3) just relation; (4) neither. We think this makes it quite explicit what is in the data.
>
> There is some potential overlap in the answers (e.g., the meta-train data might contain the answer “US” for a question about someone’s country of origin, which could be in the test data for another location query). But again, we don’t think this can be characterized as data leakage.
>
> We conducted the following analysis that analyzes how much predicted answer appears in a meta-training set. We see that it’s not memorization of answers in the meta-training set.
>
> |                   % Predicted answer Appear in Meta-Training set                  | In-Domain | OOD (Entity) | OOD (Relation) | OOD (Both) |
> |:---------------------------------------------------------------------------------:|:---------:|:------------:|:--------------:|:----------:|
> | Gold Answer                                                                       |    100    |     44.7     |      43.2      |    24.6    |
> | Base                                                                              |    68.5   |     50.2     |       54       |    41.5    |
> | CPT                                                                               |    68.3   |     41.1     |      48.1      |    29.1    |
> | Active-Reading CPT                                                                |     43    |     27.3     |      21.3      |     9.6    |
> | Meta-Aug CPT (i.e. *train on exact test question* for In-Domain and OOD (Entity)) |    97.5   |     44.9     |      64.8      |    36.4    |
> | MEMIT (Ctrl RippleEdit)                                                           |     70    |     46.9     |      50.4      |    34.1    |
> | MEND+Propagation                                                                  |    94.9   |     64.1     |       46       |    26.6    |
>
>
> > Why was naive continued pretraining chosen as the main baseline instead of more competitive and parameter-efficient alternatives, such as fine-tuning plus model merging?
>
> We only find [1] that uses model merging for knowledge editing. We will compare this baseline in the future. This paper was a recent paper that we weren’t aware of when submitting. Thanks for the catch.
>
> [1] Model Merging for Knowledge Editing ACL-Industry 2025; Zichuan Fu, Xian Wu, Guojing Li, Yingying Zhang, Yefeng Zheng, Tianshi Ming, Yejing Wang, Wanyu Wang, Xiangyu Zhao

---

### Official Review · Reviewer_NUWd · 2025-10-31

**Soundness:** 3
**Presentation:** 3
**Contribution:** 3
**Rating:** 6
**Confidence:** 3

**Summary:**

The proposal addresses a limitation of LLM editing: poor knowledge propagation after factual edits. Building on MEND, the authors propose a propagation-aware hypernetwork that learns edits for locality (updating a fact) and also for reasoning consistency on related facts. The approach modifies MEND’s meta-training objective—rather than training the hypernetwork to reproduce paraphrases. The model takes the gradient of a factual edit, processes it through the hypernetwork to generate a low-rank update, and applies this to the base model.

Experiments on RippleEdit and the proposed Controlled RippleEdit dataset show better generalization and multi-hop reasoning compared to ROME, MEMIT, MEND, and CPT. The idea is conceptually clean, empirically supported, and practically relevant, though limitations remain in generalization and analysis depth.

**Strengths:**

The paper identifies a real and concrete failure mode in current editing methods — factual edits that do not generalize to reasoning queries. This is not a contrived setting; it mirrors issues observed in ROME and MEMIT where edits fix a statement but fail to propagate logically. The authors’ framing of “propagation” as an optimization target is clearly motivated.

The proposed modification of MEND is meaningful without major lift: replacing paraphrase-based meta-objectives with reasoning-oriented loss functions. The hypernetwork’s gradient-to-weight mapping is trained end-to-end to encourage edits that also benefit multi-hop inference.

The evaluation propagation across single-hop, multi-hop, and out-of-domain splits with  baseline set (ROME, MEMIT, MEND, CPT) covers the key spectrum with clear metrics and ablations is well documented. Beyond benchmarks, this work hints at a general principle with real potential for continual knowledge maintenance in production-scale LLMs.

**Weaknesses:**

The claims on Controlled RippleEdit are good, It’s unclear whether the same gains would hold in more natural open-domain QA or reasoning settings where there isn't a simplified reasoning path entity -> relation -> derived fact.

The method is only tested on small models (~1B). The hypernetwork’s compute overhead and stability when editing larger-scale LLMs (7B–70B) are not discussed. Since hypernetworks are known to scale poorly with model dimension, I would be curious to hear more from the authors regarding this.

While the method empirically improves reasoning accuracy, the paper does not analyze where or how propagation happens inside the model (e.g., which layers change, how edits flow through attention patterns). This understanding could strengthen the paper further making it more explanatory and empirical.

**Questions:**

Some follow up questions,
- Can you quantify the runtime and memory overhead of hypernetwork-based updates?
- Can you provide analysis of which layers absorb the propagated knowledge?
- How sensitive is performance to the balance between locality and propagation loss weights?

---

> ### Author Response · Authors · 2025-11-23
> **Response to reviewer**
>
> > Whether the same gain would hold in more natural open-domain QA
>
> We agree it would be great to see results in more natural open domain QA datasets, yet there are no such datasets available. We think it would be challenging to generalize, as most editing methods make very targeted updates and the model is not ready to resurface the knowledge flexibly [1].
>
> [1] RuleEdit: Benchmarking Rule-Level Knowledge Editing in Large Language Models; ICLR 2026 under review
>
>
> > only tested on 1B model.
>
> Thanks for the suggestion, we agree showing generalization to larger models would be helpful. We also test on the 3B model in Table 28. In the table below, we also present new results with Llama-3.2-8B-base-QA and show our method is still effective to enable the model to answer propagation questions (single edit) in OOD (Entity) and OOD (Relation).
>
> Due to time and budget constraints, we use LoRA (`all-linear`) for CPT baselines and target 2 MLP layers (we edited 12 layers for 1B model) for our “MEND+Propagation” method. For LoRA, we adjust the learning rate so that it reaches a similar loss as Full Finetuning.
>
> |                                  |        In-Domain        |             |       OOD (Entity)      |             |     OOD (Relation)     |             |       OOD (Both)       |             |
> |----------------------------------|:-----------------------:|:-----------:|:-----------------------:|:-----------:|:----------------------:|:-----------:|:----------------------:|:-----------:|
> | Llama-3.2-8B-base-QA Single Edit | Overall (2284) Accuracy | Specificity | Overall (1368) Accuracy | Specificity | Overall (421) Accuracy | Specificity | Overall (447) Accuracy | Specificity |
> | Base                             |           6.8           |     91.6    |           6.3           |     89.5    |          10.3          |     93.4    |           9.9          |     92.5    |
> | Prepend                          |           74.7          |     92.4    |           73.2          |      90     |          73.8          |     91.1    |          67.5          |      94     |
> | CPT (LoRA)                       |           15.4          |     90.4    |           13.8          |     88.3    |          16.7          |      92     |          16.6          |     89.2    |
> | Active-Reading CPT (LoRA)        |           19.6          |     80.2    |           17.7          |     81.3    |          22.4          |     79.4    |        **19.6**        |     76.7    |
> | MEND+Propagation                 |         **65.2**        |     92.9    |          **41**         |     87.8    |        **30.4**        |     90.7    |          15.4          |     87.9    |
>
>
> We see a similar performance pattern to smaller models, that our method can answer propagation questions (single edit) in OOD (Entity) and OOD (Relation). Note we view Prepend as a method that adds context, but is not a baseline.
>
>
> > The hypernetwork’s compute overhead and stability when editing larger-scale LLMs (7B–70B) are not discussed.
>
> It would be challenging to scale to larger sizes. To work within similar resource constraints, we need to make other modifications such as editing fewer layers. Other engineering improvement on the codebase and might need a new design of the hypernetwork. We view that the engineering effort does not undermine our scientific contribution; and we view potential design of hypernetwork as an orthogonal direction for future followup but out of scope for our work.
>
>
> > Can you quantify the runtime and memory overhead of hypernetwork-based updates?
>
> We report profiling in Table 16 and 17 (Line 1284 --1308), showing that our method is cost-effective.
>
> > Can you provide analysis of which layers absorb the propagated knowledge?
>
> Our ablation study Table 13 and 14 (Line 1230-1255) shows that the mid-upper layer is more effective for propagation.
>
> > How sensitive is performance to the balance between locality and propagation loss weights?
>
> Our hyperparameter choice is adopted from choices in MEND and in preliminary study we didn’t find making small changes to coefficient affects performance much.

---

### Official Review · Reviewer_2CgZ · 2025-11-01

**Soundness:** 3
**Presentation:** 2
**Contribution:** 2
**Rating:** 4
**Confidence:** 3

**Summary:**

This paper addresses a critical and recognized gap in knowledge editing: ensuring that injected facts are not just memorized but can also be used for reasoning (i.e., knowledge propagation). The authors' core proposal is an intuitive fix to hypernetwork-based editors (like MEND): they align the meta-training objective with this goal by optimizing the hypernetwork to solve propagation questions directly, rather than simple paraphrases. A key contribution is the introduction of a new synthetic benchmark, 'Controlled RippleEdit,' which is designed to evaluate multi-hop reasoning and out-of-domain (OOD) generalization more effectively. The results are promising, especially on their new dataset, but show limitations in multi-edit and OOD settings.

**Strengths:**

**Clear Problem Definition**: The paper clearly defines and targets a crucial, practical limitation of current knowledge editing methods: their failure to propagate knowledge, where models can parrot a new fact but cannot reason with it. This is a well-motivated and important problem for the field.

**Simple and Effective Solution**: The proposed solution is both simple and effective. By modifying the hypernetwork's outer-loop objective to use propagation questions, the authors align the meta-training process directly with the desired outcome. This intuitive change yields significant performance gains, particularly on the challenging non-verbatim questions in RippleEdit, where it achieves nearly 2x the accuracy of the next-best system.

**Valuable Benchmark Contribution**: A significant contribution is the new 'Controlled RippleEdit' dataset. This benchmark is well-designed, using fictional entities linked to real-world knowledge to create a controlled testbed for multi-hop reasoning . Its inclusion of OOD splits (for both entities and relations) is particularly valuable for assessing whether the editor has learned a generalizable propagation mechanism or is merely overfitting to seen relations

**Weaknesses:**

1. **Collapse in Multi-Edit Performance**: The method's performance in the multi-edit setting is a major concern. While MEND+Propagation is a strong performer for a single edit (Table 2) , its multi-edit counterpart (RLEdit+Propagation) shows a severe collapse in efficacy, dropping from 76.7 (1 edit) to 48.6 (10 edits) and 29.9 (20 edits). In stark contrast, the Meta-Aug CPT baseline remains highly effective (88.6 for 10 edits, 89.2 for 20 edits). This rapid degradation suggests the hypernetwork approach, when combined with a propagation objective, does not scale sequentially. Why does this approach degrade so much faster than CPT? The paper lacks a sufficient analysis of this critical failure mode.

2. **Comparison to 'Prepend' Baseline**: The claims regarding the RippleEdit results seem overstated. The abstract highlights a 'nearly 2x' accuracy gain , but Table 1 shows that on the key 'Efficacy Non-Verbatim' metric, the proposed MEND+Propagation (22.4) achieves an identical score to the 'Prepend' (in-context learning) baseline (22.4). While this is a significant improvement over other parametric methods like MEMIT (12.7), it also implies that the complex meta-training provides no practical advantage over simple in-context learning on this dataset. While the method does outperform Prepend on the new Controlled RippleEdit dataset (76.7 vs 38.1), the paper should more clearly discuss whether this gap justifies the significant overhead (training, extra parameters, latency) of a hypernetwork versus the zero-cost 'Prepend' baseline.
3. **Underdeveloped Analysis of OOD Failure**: The analysis of the OOD generalization failure is underdeveloped. The paper candidly reports a significant performance drop from in-domain (76.7) to OOD (Both) (18.3). However, it offers little insight into why this failure occurs. The goal of meta-learning a hypernetwork is to learn a general update function. Why does this function fail so completely when encountering unseen relations or entities? Is the network simply overfitting to the specific relations seen during training (e.g., learning a specific mapping from a 'born in' gradient to 'capital of' weights) rather than learning a truly generalizable reasoning mechanism? A more in-depth analysis of these failures would be necessary to understand what the model has actually learned.

**Questions:**

See weaknesses

---

> ### Author Response · Authors · 2025-11-23
> **Response to reviewer (1/2)**
>
> > Collapse in Multi-edit
>
> We would like to clarify that the Meta-Aug CPT system, by design, generates the *exact test questions* in the In-domain and Out-of-domain (Entity) settings. That’s partially why it shows such strong performance in the In-Domain and OOD(Entity) settings, while the gap between our method and Meta-Aug CPT in other OOD settings decreases significantly.  And we strongly disagree that “88.6 for 10 edits, 89.2 for 20 edits” is evidence for Meta-Aug CPT being effective because this is a specificity measure (i.e. whether model crushes after edit), not efficacy measure (i.e. whether editing succeeds).
>
> We view the performance of non-augmentation methods as relevant: these methods don’t incur extra cost for training at test time. Multi-edit is a challenging setting for all models,  even for Adaptive Reading where the training corpus is 95x more tokens.
>
>
> > Comparison to Prepend baseline:
>
> Our study focuses on parametric updating and injection of knowledge, but Prepend or any in-context learning baseline does not produce parametric update and are not editing methods (Line 247). We therefore view Prepend as a method that adds context, not as a baseline. This follows the same dichotomy of methods in prior work [1]. Our claim of 2x performance is comparing MEND+Propagation (22.4) with the best-performing *parametric* baseline MEMIT (12.7). We will clarify this in the revised paper.
>
> [1] Propagating Knowledge Updates to LMs Through Distillation NeurIPS 2023
> Shankar Padmanabhan, Yasumasa Onoe, Michael J.Q. Zhang, Greg Durrett, Eunsol Choi

---

> ### Author Response · Authors · 2025-11-23
> **Response to reviewer (2/2)**
>
> > Under-developed analysis of OOD failure
>
> Thank you for your suggestion, we agree OOD failure analysis would be very helpful! We conduct two additional analyses and report them below.
>
> **A. Analysis on experimental results on two subsets of the dataset.**
>
> We report results on two subset of evaluation data on OOD (Entity), whether the gold answer appears verbatim in the meta-training set.
> |                       Efficacy                       | In-Domain               | OOD (Entity)                             |                                                  |                         |
> |:----------------------------------------------------:|-------------------------|------------------------------------------|--------------------------------------------------|-------------------------|
> |         Llama-3.2-1B-base-QA-FMT Single Edit         | Overall (2284) Accuracy | Gold answer in meta-train (612) Accuracy | Gold answer **not** in meta-train (756) Accuracy | Overall (1368) Accuracy |
> | Base                                                 |           8.3           |                    13                    |                        2.4                       |           7.1           |
> | CPT                                                  |           18.1          |                   22.2                   |                       12.8                       |            17           |
> | Active-Reading CPT                                   |           19.6          |                   22.1                   |                       16.8                       |           19.1          |
> | Meta-Aug CPT (i.e. **train on exact test question**  for In-Domain and OOD (Entity)) |           80.3          | 79.6                                     | 78.8                                             | 79.1                    |
> | MEMIT (Ctrl RippleEdit)                              |           12.0          |                   19.9                   |                        7.9                       |           14.4          |
> | MEND+Propagation                                     |           76.7          |                   49.1                   |                        24                        |           35.2          |
>
> We observe:
> 1. The performance on “Gold answer not in meta-train” subset is lower than “Gold answer in meta-train”, yet performance is nontrivial (24). If hypernetwork learns by pure memorization, this accuracy would be closer to 0.
> 2. “Gold answer in meta-train” subset has lower performance (compared to in-domain performance), and this means the hypernetwork might have problems identifying related knowledge (e.g., currency_of for an unseen entity) for an OOD entity. Since knowledge of OOD entities is not activated during meta-learning, such generalization is still encouraging.
>
> Overall, we think our analysis implies that the propagation (in OOD settings) is not targeted enough likely because the knowledge of unseen relations or entities are not activated when learning the hypernetwork.
>
>
> **B. Analysis on the evaluation data themselves.**
>
> We investigated the problem of whether the answers on OOD questions appear verbatim in the meta-training set; and we show in the table below that the predicted answer is not always verbatim in the meta-training set.
>
> |                   % Predicted answer Appear in Meta-Training set                  | In-Domain | OOD (Entity) | OOD (Relation) | OOD (Both) |
> |:---------------------------------------------------------------------------------:|:---------:|:------------:|:--------------:|:----------:|
> | Gold Answer                                                                       |    100    |     44.7     |      43.2      |    24.6    |
> | Base                                                                              |    68.5   |     50.2     |       54       |    41.5    |
> | CPT                                                                               |    68.3   |     41.1     |      48.1      |    29.1    |
> | Active-Reading CPT                                                                |     43    |     27.3     |      21.3      |     9.6    |
> | Meta-Aug CPT (i.e. **train on exact test question** for In-Domain and OOD (Entity)) |    97.5   |     44.9     |      64.8      |    36.4    |
> | MEMIT (Ctrl RippleEdit)                                                           |     70    |     46.9     |      50.4      |    34.1    |
> | MEND+Propagation                                                                  |    94.9   |     64.1     |       46       |    26.6    |

---

### Official Review · Reviewer_kyQR · 2025-11-01

**Soundness:** 2
**Presentation:** 2
**Contribution:** 2
**Rating:** 4
**Confidence:** 4

**Summary:**

This paper studies knowledge propagation in hypernetwork-based knowledge editing for LLMs. The authors identify that vanilla MEND does not effectively propagate injected knowledge and propose training the hypernetwork with propagation-oriented objectives. Experiments on the RippleEdit dataset and a newly introduced synthetic benchmark, Controlled RippleEdit, show that the proposed method improves multi-hop knowledge propagation, particularly on non-verbatim cases.

**Strengths:**

**Clear empirical focus.** Evaluates on RippleEdit and introduces a controlled synthetic benchmark to isolate propagation effects.

**Simple and practical modification.** Adapting hypernetwork training to include propagation supervision is understandable and lightweight, making it potentially attractive for practitioners.

**Improved non-verbatim propagation performance.** Shows strong gains on non-verbatim multi-hop queries where existing methods struggle (e.g., 2× accuracy improvement reported).

**Weaknesses:**

1. What exactly is “knowledge propagation” in model editing? Although prior works (e.g., RippleEdit) have discussed knowledge propagation, this paper, as an independent contribution, should clearly define such an important and non-common concept in both the introduction and background sections. However, the manuscript only briefly mentions in Line 37 that existing model editing methods struggle to achieve knowledge propagation, without providing further explanation or analysis. This lack of definition and conceptual clarity is confusing.

2. Limited contribution beyond MEND. The method is built on MEND, but the overall contribution appears weak. My understanding is that the authors modify MEND to make it applicable to knowledge propagation tasks. However, it remains unclear whether this modification strategy can generalize to other hypernetwork-based editing methods, as the paper does not discuss or evaluate such cases. Therefore, I have concerns regarding the significance and generality of the contribution.

3. Lack of theoretical support for intuitive design choices. The proposed method involves several intuitive design choices, but lacks theoretical analysis. I am particularly interested in the mechanism underlying why such a simple adjustment can notably improve MEND’s performance on knowledge propagation tasks. Without such analysis, the method feels heuristic rather than principled.

4. Missing recent baselines. Model editing is a rapidly evolving field with many recent competitive baselines, including but not limited to AlphaEdit [1], RLEdit [2], and DAFNet [3]. However, these methods are not included in the experiments. In addition, the newly proposed Controlled RippleEdit dataset requires stronger empirical validation with recent baselines to demonstrate fairness and credibility. Given the current scale of horizontal comparison, I have reservations regarding the effectiveness of the proposed method and dataset.

Reference

[1] Fang J, Jiang H, Wang K, et al. AlphaEdit: Null-Space Constrained Knowledge Editing for Language Models[C]//The Thirteenth International Conference on Learning Representations.

[2] Li Z, Jiang H, Chen H, et al. Reinforced Lifelong Editing for Language Models[C]//Forty-second International Conference on Machine Learning.

[3] Zhang T, Chen Q, Li D, et al. DAFNet: Dynamic Auxiliary Fusion for Sequential Model Editing in Large Language Models[C]//Findings of the Association for Computational Linguistics ACL 2024. 2024: 1588-1602.

**Questions:**

Please see weakness

---

> ### Author Response · Authors · 2025-11-23
> **Response to reviewer (1/2)**
>
> > What exactly is “knowledge propagation” in model editing?...
>
> We acknowledge there is no “formal” definition for knowledge propagation and we follow the intuitive definition of prior work [1,2]. Knowledge propagation means after editing new knowledge into the model, the updated model can not only reproduce this knowledge, but make inferences based on it.
>
> We have concrete examples in Line 101 and Line 106-107. We write here for convenience:
> ```
> Injected fact: “Keir Starmer was elected prime minister of the UK”
> Propagation question: Q: What year was the prime minister of the UK born? A: 1962;
> ```
>
> While this is a desirable property for LLMs, it is difficult to define formally because propagating knowledge to the complete deductive closure of a fact is impractical (see the logical omniscience problem [3]). We will update the paper to include this discussion.
>
> [1] Propagating Knowledge Updates to LMs Through Distillation NeurIPS 2023
> Shankar Padmanabhan, Yasumasa Onoe, Michael J.Q. Zhang, Greg Durrett, Eunsol Choi
>
> [2] Evaluating the Ripple Effects of Knowledge Editing in Language Models TACL 2024
> Roi Cohen, Eden Biran, Ori Yoran, Amir Globerson, Mor Geva
>
> [3] Dealing with logical omniscience: Expressiveness and pragmatics. Artificial Intelligence, 175(1), 220-235 Halpern, J. Y., & Pucella, R. (2011).
>
>
> > Limited contribution beyond MEND. However, it remains unclear whether this modification strategy can generalize to other hypernetwork-based editing methods, as the paper does not discuss or evaluate such cases.
>
> We do not only modify MEND. In Line 162-176, we experimented with RLEdit [1], a newer editing method that targets multi-fact and multi-turn edit scenarios. Our results in Table 3 shows that our modification applies to RLEdit as well. Our modification can work for any editing method that trains a hypernetwork with a supervised finetuning objective.
>
> [1] Reinforced Lifelong Editing for Language Models ICML 2025;  Zherui Li, Houcheng Jiang, Hao Chen, Baolong Bi, Zhenhong Zhou, Fei Sun, Junfeng Fang, Xiang Wang
>
>
> > Lack of theoretical support for intuitive design choices.
>
> Our intuition is that incorporating propagation as a criterion in meta-learning will drive the hypernetwork to achieve propagation. The key empirical questions are twofold. First, is the hypernetwork capable of representing propagation? The answer our work provides is yes: while the mechanisms of knowledge storage in Transformers are not possible to precisely pin down, past work like ROME and “Transformers Feedforward Layers are Key-Value Memories” posit mechanisms for knowledge storage which are compatible with this approach. Second, does the hypernetwork generalize? This is the question that most of our experiments seek to address. As we mentioned in our main paper (Line 412), such intuition is supported by our additional analysis (Table 14 and 19; see Line 994).
>
> We acknowledge the lack of theoretical analysis, but do not view this as a critical weakness. Establishing theoretical analysis for large-scaled LLMs handling real world text is challenging, and empirical studies should be valued on its own.

---

> ### Author Response · Authors · 2025-11-23
> **Response to reviewer (2/2)**
>
> > Missing recent baselines:
>
> Thank you for the suggestions! The suggested related work focuses on increasing the number of edits or number of edit turns, which diverges with our main goal. For example, AlphaEdit’s performance on MQuake shows only negligible improvement over MEMIT in single-edit scenarios [1].
>
> We report the result of AlphaEdit on Controlled RE (our benchmark). We find AlphaEdit archives **worse performance** than MEMIT, which is not surprising as propagation was not their main goal. We will include this baseline and discuss them in the paper.
>
>  |      Llama-3.2-1B-QA      | In-Domain |             | Out-of-Domain (Entity) |             | Out-of-Domain (Relation) |             | Out-of-Domain (Both) |             |
> |:-------------------------:|:---------:|:-----------:|:----------------------:|:-----------:|:------------------------:|:-----------:|:--------------------:|:-----------:|
> |           Method          |  Efficacy | Specificity |        Efficacy        | Specificity |         Efficacy         | Specificity |       Efficacy       | Specificity |
> |            Base           |    8.3    |     94.7    |           7.1          |     94.3    |            8.9           |     94.2    |         10.9         |     90.7    |
> |     MEMIT (Wikipedia)     |    12.8   |     94.4    |          14.4          |     94.4    |           12.0           |     93.9    |         13.8         |     90.0    |
> |   MEMIT (Controlled RE)   |    12.0   |     94.6    |          13.3          |     94.5    |           11.1           |     94.3    |         11.6         |     90.2    |
> | AlphaEdit (Wikipedia)     | 11.3      | 94.3        | 12.7                   | 94.4        | 13.7                     | 93.8        | 13.7                 | 90.1        |
> | AlphaEdit (Controlled RE) | 10.9      | 94.3        | 11.3                   | 94.4        | 13.8                     |  94.0       | 13.3                 | 89.7        |
> | MEND+Propagation          | **76.7**  | 95.5        | **35.2**               | 81.6        | **34.5**                 | 84.0        | **18.3**             | 77.5        |
>
>
> [1] Fang J, Jiang H, Wang K, et al. AlphaEdit: Null-Space Constrained Knowledge Editing for Language Models[C]//The Thirteenth International Conference on Learning Representations.
>
>
> > In addition, the newly proposed Controlled RippleEdit dataset requires stronger empirical validation with recent baselines to demonstrate fairness and credibility.
>
> Based on the construction procedure of this dataset, we believe it has the following properties:
> 1. Unique answer for each question (up to paraphrases which are captured by LLM-as-a-judge)
> 2. Answers can be determined based on the given edit context
>
> The task has a different focus (with design decisions discussed in the paper), different from those of prior work, but we don’t quite understand how fairness and credibility would be assessed here.

---

### Note · Authors · 2025-12-30

I have read and agree with the venue's withdrawal policy on behalf of myself and my co-authors.